# Urokinase-Type Plasminogen Activator Receptor (uPAR) in Inflammation and Disease: A Unique Inflammatory Pathway Activator

**DOI:** 10.3390/biomedicines12061167

**Published:** 2024-05-24

**Authors:** Mostafa Hamada, Kyle Steven Varkoly, Omer Riyadh, Roxana Beladi, Ganesh Munuswamy-Ramanujam, Alan Rawls, Jeanne Wilson-Rawls, Hao Chen, Grant McFadden, Alexandra R. Lucas

**Affiliations:** 1College of Medicine, Kansas City University, 1750 Independence Ave, Kansas City, MO 64106, USA; mostafa.hamada@kansascity.edu (M.H.); omer.riyadh@kansascity.edu (O.R.); 2Department of Internal Medicine, McLaren Macomb Hospital, Michigan State University College of Human Medicine, 1000 Harrington St., Mt Clemens, MI 48043, USA; 3Department of Neurosurgery, Ascension Providence Hospital, Michigan State University College of Human Medicine, 16001 W Nine Mile Rd, Southfield, MI 48075, USA; roxanabeladi@gmail.com; 4Molecular Biology and Immunobiology Division, Interdisciplinary Institute of Indian System of Medicine, SRM Institute of Science and Technology, Kattankulathur 603203, India; mrganesh2000@hotmail.com; 5School of Life Sciences, Arizona State University, 427 E Tyler Mall, Tempe, AZ 85281, USA; alan.rawls@asu.edu (A.R.); jeanne.wilson-rawls@asu.edu (J.W.-R.); 6Department of Tumor Center, Lanzhou University Second Hospital, Lanzhou 730030, China; chenhao3996913@163.com; 7Center for Personalized Diagnostics, Biodesign Institute, Arizona State University, 727 E Tyler St., Tempe, AZ 85287, USA; grantmcf@asu.edu

**Keywords:** urokinase-type plasminogen receptor, uPAR, inflammation, cancer, sepsis, virus, inflammatory bowel disease, Serp-1, serpin

## Abstract

The urokinase-type plasminogen activator receptor (uPAR) is a unique protease binding receptor, now recognized as a key regulator of inflammation. Initially, uPA/uPAR was considered thrombolytic (clot-dissolving); however, recent studies have demonstrated its predominant immunomodulatory functions in inflammation and cancer. The uPA/uPAR complex has a multifaceted central role in both normal physiological and also pathological responses. uPAR is expressed as a glycophosphatidylinositol (GPI)-linked receptor interacting with vitronectin, integrins, G protein-coupled receptors, and growth factor receptors within a large lipid raft. Through protein-to-protein interactions, cell surface uPAR modulates intracellular signaling, altering cellular adhesion and migration. The uPA/uPAR also modifies extracellular activity, activating plasminogen to form plasmin, which breaks down fibrin, dissolving clots and activating matrix metalloproteinases that lyse connective tissue, allowing immune and cancer cell invasion and releasing growth factors. uPAR is now recognized as a biomarker for inflammatory diseases and cancer; uPAR and soluble uPAR fragments (suPAR) are increased in viral sepsis (COVID-19), inflammatory bowel disease, and metastasis. Here, we provide a comprehensive overview of the structure, function, and current studies examining uPAR and suPAR as diagnostic markers and therapeutic targets. Understanding uPAR is central to developing diagnostic markers and the ongoing development of antibody, small-molecule, nanogel, and virus-derived immune-modulating treatments that target uPAR.

## 1. Introduction



*Hidden strength*

*Strange power*

*Pervasive*

*Invasive*

*Universal and unknown*

*Life to life.*

*Alexandra Lucas, MD, FRCP(C)—March 2024*



### 1.1. The Thrombolytic Plasminogen Activator (PA) Protease Cascade

The primary physiological role of the urokinase-type plasminogen activator (uPA) system was first identified as a thrombolytic or clot-dissolving protease cascade. The binding of uPA to urokinase-type plasminogen activator receptor (uPAR) initiates a cascade of events, converting plasminogen into plasmin, which then drives fibrinolysis. Through activation of plasmin the uPA/uPAR complex can also initiate cell activation, tissue remodeling, and angiogenesis. Here, we begin with a discussion of the role of uPA in fibrinolysis and clot breakdown.

The tissue- and urokinase-type plasminogen activators (tPA and uPA, respectively) are the two serine protease enzymes known to activate plasmin and break down clots. tPA and uPA have differing cell surface receptors [1,2,3]. Both tPA and uPA have been used as thrombolytic agents for myocardial infarction and other acute thrombotic events such as strokes (cerebrovascular accident), deep venous thrombosis (DVT), and pulmonary embolism. In the vascular system, although tPA and uPA both activate plasmin to cleave fibrin, tPA is the principal thrombolytic or clot-dissolving protease in the circulating blood. uPA also has clot-dissolving activity but is now reported to function predominantly as an inflammatory mediator.

Plasminogen is cleaved to form plasmin, which is also a serine protease with a diverse set of functions, in addition to clot lysis. Plasmin activates pro-matrix metalloproteinases (MMPs) that drive extracellular matrix degradation and also release and activate latent tissue growth factors, such as transforming growth factor-beta (TGF-β) [1,2]. Inhibitors of tPA, uPA, and plasmin function to regulate these pathways. Serine protease inhibitors, termed *serpins*, regulate the thrombolytic and thrombotic pathways and have been extensively studied. Mammalian serpins that regulate the uPA, tPA, and plasmin pathways include plasminogen activator inhibitor-1 (PAI-1, SERPINE1), PAI-2 (SERPINB2) [1,3], and other serpins, specifically, α2-antiplasmin (SERPINF2) and α2-macroglobulin [3,4]. PAI-1 and PAI-2 are active when binding to the uPA/uPA receptor complex (uPA/uPAR), but antiplasmin is most active in targeting proteases in the circulation. 

The cell-associated uPA/uPAR complex and associated soluble uPAR cleavage products (suPAR) have now been identified as markers for severe immune and inflammatory diseases, as well as markers for cancer. 

### 1.2. Urokinase-Type Plasminogen Activator (uPA)

The uPA was first identified as a thrombolytic. However, as noted above, this unique serine protease also has a leading role in the inflammatory response, with a wide range of physiologically significant functions [5,6,7,8,9,10,11,12,13,14,15]. 

Historically, the uPA was first identified and reported in 1947 as a thrombolytic by MacFarlane and Pilling, with the seminal identification of uPA as a protein in urine [5]. Approximately five years later, Sobel and colleagues officially christened this protein “urokinase” [6], now used as a thrombolytic. Urokinase, or uPA, is in various bodily fluids including plasma, seminal fluid, and extracellular matrix (ECM) [7]. uPA is also detected on many immune cells and in many cancers. Thus, uPA, also termed urokinase, was first studied as a thrombolytic and only later found to have widespread immune pathway functions.

uPA is synthesized and secreted as a single glycosylated proenzyme or zymogen, referred to as pro-uPA, which consists of 411 amino acids, with little or no intrinsic enzymatic activity. Pro-uPA contains three distinct domains: a growth factor domain, sharing structural similarities with the epidermal growth factor (GFD); a kringle domain (KD); and a serine protease domain [8,14,15,16,17] (Figure 1). The GFD and KD are situated at the N-terminus, whereas the catalytic serine protease domain (amino acids 159–411) is located at the C-terminus. A linker region connects the N- and C-terminal regions [9]. 

Once secreted, pro-uPA undergoes cleavage at the peptide bond between Lys158 and IIe159 to generate the double chain form of uPA. The two-chain form of uPA (tcuPA) is still linked through a disulfide bond [10]. Thus, a key event in the activation of pro-uPA is the cleavage of a specific peptide bond within the pro-uPA molecule. tcuPA consists of a heavy chain (A-chain) and a light chain (B-chain). The structural variations in uPA contribute to the functional versatility of uPA. tcuPA and the single-chain form (scuPA) can both bind to uPAR; however, tcuPA is 250 times more potent at generating plasmin than scuPA [11]. A range of different proteases, such as thrombin and elastase, have also been identified as agents that facilitate cleavage of pro-uPA, resulting in the generation of active high-molecular-weight uPA [10]. Additionally, various other activating proteins including cathepsin B and L, mast cell tryptase, and nerve growth factor-gamma, have been recognized as capable of activating pro-uPA [12,13,14]. Among these, plasmin stands out as the most efficient enzyme for the conversion of pro-uPA into its active counterpart, uPA [10]. Plasmin formed from plasminogen by uPA activation, thus, activates uPA. 

### 1.3. Urokinase-Type Plasminogen Activator Receptor (uPAR)

The urokinase-type plasminogen receptor (uPAR), also termed CD87 and encoded by the PLAUR gene, is now recognized as a central immune regulating receptor. uPAR is a 50–60 kDa glycosylated protein involved in multiple physiological and pathological processes [7,17,18,19,20,21,22,23,24,25,26,27,28,29]. uPAR is part of the Ly6 (lymphocyte antigen-6)/uPAR family of receptors. Ly6/uPAR encodes proteins containing at least one conserved functional motif, termed the LY6/uPAR (LU) domain [18,19]. The LY6/uPAR superfamily of protein receptors were first identified in T lymphocytes and comprise over 30 genes found in insects, fish, amphibians, reptiles, birds, and mammals [18]. uPAR was first discovered by three researchers simultaneously in 1985: Vassalli, Del Roso, and Blasi [21,22,23].

The uPAR is expressed on the surface of endothelial cells, neutrophils, monocytes/macrophages, T cells, and fibroblasts, as well as many cancers, with reported major roles in inflammatory and cancer cell responses (Figure 2). uPAR can alter both intracellular signaling and cell activation, as well as extracellular adhesion and migration. uPAR binds vitronectin and is associated with a large lipid raft of membrane proteins that include integrins, growth factors including transforming growth factor beta (TGFβ), fibroblast growth factor (FGF), and vascular endothelial growth factor (VEGF), as well as chemokine receptors (i.e., CXCR4) and low-density lipoprotein receptor-related protein (LRP). uPA bound to uPAR (uPA/uPAR), as noted, also activates extracellular-matrix-degrading enzymes, such as the MMPs, and can activate growth factors [7,17,18,19].

uPAR is a single-chain polypeptide, further subdivided into three domains (D1, D2, and D3), each approximately 90 amino acids [7]. Following cleavage of this single-chain polypeptide, the D1 domain binds uPA (Figure 2). Although uPAR serves as the attachment site for uPA to the cell surface, uPAR is not a transmembrane protein, but is instead a glycosylphosphatidylinositol (GPI)-linked receptor (Figure 2). uPAR is covalently bonded to the GPI, a glycophospholipid in the outer leaflet of the membrane. uPAR is reported to alter internal cellular activity via interacting with adjacent transmembrane proteins in this large lipid raft of associated membrane proteins, such as the integrins [7,17,18]. uPAR can also alter cellular activation when incorporated into a cell, as is seen when uPAR is internalized after binding to uPA and the plasminogen activator inhibitor-1 (PAI-1).

The binding of uPA to uPAR is an intricate and tightly regulated molecular interaction with a pivotal role in various physiological and pathological processes. The uPA/uPAR complex is characterized by high-affinity binding, which is achieved through specific binding of three consecutive binding domains on both uPA and uPAR. Crystal structure analysis has provided a detailed uPA/uPAR binding structure. The Ly6/uPAR family of receptors maintains a three-finger structure. The three-dimensional structure of uPAR forms a central hydrophobic site deep in its D1 domain core structure. This location serves as the ligand-binding site for uPA, specifically the β-hairpin structure on the surface of the growth-factor-like domain (GFD) [19,20]. This tight binding near the center of uPAR enables the external surfaces of uPA to interact with other proteins in the lipid raft where uPAR sits, including proteins such as vitronectin (Vn). Vn is a multifunctional glycoprotein present in both the extracellular matrix and blood and is closely involved in integrin interactions and cell adhesion. uPAR can interact with uPA, plasminogen, glycosaminoglycans, and collagen, and when bound to uPA can stabilize interactions with the inhibitor PAI-1, a mammalian serpin. Such multifaceted binding allows uPAR to regulate other activities such as proteolysis and cell adhesion, including, for example, formation of actin-rich lamellipodia on Vn [24,25]. This molecular model provides a basis for the potential use and development of small-molecule inhibitors of uPA/uPAR interactions, a growing field, as discussed in this review.

uPAR, together with Vn, is closely associated with transmembrane integrin proteins that are associated with cell migration, adhesion, and proliferation. uPAR interacts with many integrins; however, uPAR has the highest affinity for the fibronectin receptors α5β1 and α3β1 (Figure 3). Other integrins that interact with uPAR include macrophage 1 antigen (Mac1, a complement receptor also known as CD11b/CD18 or CR3), a leukocyte αMβ2 -integrin receptor that binds fibrinogen and the αvβ5, and the αvβ3 integrins that bind Vn [25]. Through uPAR’s interaction with integrins and other associated receptors including G protein-coupled receptors (GPCRs) in the uPAR lipid raft, cellular signaling, cellular activation, and cellular migration are modified. GPCRs include the chemokine receptors that are central to cellular chemotaxis and migration, linking chemokines bound to glycosaminoglycans in the cellular matrix to the cell surface-linked GPCRs, forming chemotactic cell gradients. One such receptor includes uPAR-agonization of the FPRL1/LXA4R GPRC necessary for uPA chemotactic activity, resultantly linking the fibrinolytic and inflammatory responses in the process [30]. Once PAI-1 binds to the uPA/uPAR complex, the inhibited complex is internalized, further modifying intracellular signaling and activation [24,25,26].

Through close interactions, the activation of integrins can further modulate cellular signaling and increase production of pro-inflammatory cytokines such as interleukin-1 beta (IL-1β), IL-6, etc. Additionally, the induction of JAK-STAT by integrins and other signaling pathways can increase cell-mediated immunity and division (Figure 3).

To ensure precise control, uPA/uPAR interactions are strictly regulated by various inhibitors. PAI-1 (SERPINE1 gene) is a serine protease inhibitor, a serpin that is an essential regulator for both uPA and tPA [31], preventing uncontrolled proteolysis and fibrinolysis. The balance between uPA, uPAR, and PAI-1 maintains homeostasis. uPA activity is also diminished by other cellular mechanisms, including endocytosis, when uPA/uPAR is bound to PAI-1 [32].

For uPAR to effectively bind its ligand with the highest affinity, full involvement of all three D domains of uPAR is needed. Cleavage of the D1–D2 linker domains by other proteases irreversibly blocks uPAR’s interaction with other proteins [27,28]. In alternative pathways, uPAR is converted into a soluble form that is shed from the cell surface. This soluble uPAR (suPAR) consists of all three domains which still maintain the full capacity to bind uPA and cannot be cleaved by uPA [27,29].

### 1.4. Soluble Urokinase-Type Plasminogen Activator Receptor (suPAR)

Enzymatic cleavage of the uPAR GPI anchor leads to the generation of a soluble variant of uPAR, referred to as suPAR [29,33]. This cleavage is mediated by various proteolytic enzymes, including plasmin and other proteases such as glycerophosphodiesterase (GDE3). The specific cleavage sites on uPAR may vary, but they are generally located in the extracellular region of the protein. GDE3 is a membrane-associated GPI-specific phospholipase C, that releases uPAR from the cell membrane surface. This suPAR fragment is released into the bloodstream, circulating throughout the body and making it detectable in plasma samples. suPAR retains the ability to bind to uPA, plasminogen, and other ligands, contributing to various processes involved in immune response regulation and inflammation. Increased suPAR levels in the bloodstream are associated with pathological conditions, including cardiovascular disease, renal disorders, and cancer [31]. In addition to its proteolytic activity, the binding of uPA to suPAR also triggers the same intracellular signaling pathways as uPA. suPAR can transduce signals into the cell, influencing processes such as cell proliferation, migration, and adhesion. This makes the uPA/suPAR interaction a key player in tumor and cancer progression, both as a marker for disease and potentially as a mediator of cancer and other diseases [32,34,35].

The interactions between the serpin PAI-1 and suPAR have not been fully defined but may also lead to irreversible inhibitory actions, as seen with PAI-1 inhibition of uPA. The exact function of suPAR is an area of ongoing, active research, but it is believed to be involved in innate (inflammatory) immune responses and tissue remodeling, similar to uPAR. Elevated suPAR levels are reported with increased innate and acquired immune pathway activation. suPAR in the blood can be detected through the enzyme-linked immunosorbent assay (ELISA) or other point-of-care immunoassay techniques. This technique has been studied for patients with SARS-CoV-2 infections and antibiotic administration in patients with acute SARS-CoV-2 infections (COVID-19) [36,37,38]. Additionally, suPAR levels are relatively stable throughout the day, in addition to stability in plasma samples following many freeze–thaw cycles, but can be elevated in disease [39]. suPAR has thus been proposed as a biomarker for a variety of inflammatory diseases, as well as a potential therapeutic target [40,41].

## 2. uPAR as a Marker for Severe Inflammatory Diseases and Cancer

In this section, we will begin with a full review of uPAR as a clinical marker for disease progression organized according to studies reported for individual organ systems. 

The suPAR/uPAR system has been studied as a biomarker and therapeutic target in a wide variety of diseases, from cardiovascular and hematological diseases to oncology and chronic inflammatory diseases. suPAR and uPAR have been reported as markers for systemic inflammatory response syndrome (SIRS) and sepsis, wound healing and repair, neurological diseases, infectious diseases, cancers and also renal, gastroenterology, endocrine, pulmonary, and arthritic disorders. 

The current studies examining uPAR and suPAR as markers for disease and as potential targets for new treatments in diseases associated with individual organ systems are discussed in Section 2 and listed in Table 1 by individual organ systems and diseases. 

### 2.1. Cardiovascular Disease

Cardiovascular disease is the leading cause of death in North America. Previous research has identified uPAR as a key player in the pathogenesis of cardiovascular disease, and specifically, atherosclerosis, with strong associations to leukocyte invasion and tissue remodeling (Table 1) [125]. Various cell types, such as smooth muscle cells, macrophages, and endothelial cells all express uPAR. Such discoveries have led to broad clinical studies that correlate the levels of suPAR to cardiovascular morbidity and mortality [125,126]. 

#### 2.1.1. Acute Coronary Syndrome

Acute coronary syndrome (ACS) is a medical emergency caused by thrombosis with sudden complete or partial blockage of blood flow to the heart muscle. ACS is typically caused by macrophage invasion and rupture of an atherosclerotic plaque within a coronary artery, exposing the inner atheromatous plaque, leading to platelet activation and activation of the clotting cascade, with subsequent formation of a blood clot (thrombosis). The clotting cascade is a series of serine proteases activated on the platelet surface forming a fibrin clot. The fibrin clot partially or completely blocks blood flow or can cause intermittent obstruction [127]. When blood flow is blocked this reduces oxygen supply to the heart muscle, termed ischemia. Early diagnosis and prompt treatment, often involving antiplatelet agents, anticoagulants, and revascularization procedures such as balloon angioplasty and stent implant, restore blood flow and reduce heart muscle damage, with improved survival.

In flow cytometry studies conducted on 263 angina patients, Zhang et al. found increased expression of uPAR on circulating monocytes. In those patients, elevated uPAR levels strongly correlated with other clinically used inflammatory biomarkers and clinical instability [42]. Individuals with coronary artery disease (CAD) and with elevated suPAR levels had a significantly increased all-cause mortality, including increased cardiovascular mortality [42]. Additional studies in this field have identified suPAR as a pro-inflammatory biomarker for increased risk for unstable coronary artery disease, but not for stable coronary atherosclerosis [43,44]. Increased local production of suPAR in coronary vessels is also associated with a dysfunctional endothelial cell layer in patients [45]. suPAR may allow identification of patients with a higher risk of unstable plaque and plaque rupture in coronary artery disease, potentially identifying patients who will benefit from earlier interventions, such as stent implant or bypass surgery. 

#### 2.1.2. Acute Myocardial Infarction

An acute myocardial infarction (AMI), commonly known as a heart attack, occurs when there is a complete and persistent obstruction of coronary blood flow to the heart [127,128]. The role of uPAR in the context of acute myocardial infarction (AMI) has drawn considerable attention, particularly due to its association with inflammation, leukocyte adhesion, and invasion. uPA/uPAR research is rapidly evolving in the cardiovascular field. Mouse uPA-knockout studies have shed light on the role of uPA in unstable plaque rupture and tissue infarction. One such study observed a significant reduction in myocardial infarct areas in uPA-deficient (uPA^−/−^) but not uPAR^−/−^ mice, when compared to wildtype mice, indicating a pro-inflammatory and pro-necrotic function for uPA [46]. Moreover, uPA deficiency (uPA^−/−^) protected against myocardial rupture, a complication of AMI [46]. Thus, although uPA has thrombolytic, or clot-dissolving, pathways when used as a therapeutic, in the circulation there is an apparent paradox in that reducing uPA reduces risk of unstable plaque rupture and thrombosis and ischemic myocardial damage. These studies highlight the varying functions of the uPA/uPAR complex in fibrinolysis and inflammation, with a greater role for the uPA/uPAR complex in inflammatory cell responses in unstable and inflamed atherosclerotic plaque.

In a study examining patients with AMI, uPAR expression was increased, but in stable angina patients, uPAR was not increased [47]. In this study, researchers discovered that targeting uPAR with monoclonal antibodies blocked monocytic adhesion as well as integrin-mediated fibrinogen adhesion. This study further established the pro-inflammatory role of uPAR in unstable vascular disease and indicates that uPAR is a potential target for preventing leukocyte adhesion and migration in areas of unstable plaque, arterial thrombosis, and myocardial injury.

In another study, of 1314 patients presenting to the ED with suspected MI, suPAR reliably predicted mortality after one year of presentation [48]. Further studies have linked the levels of suPAR six hours post-cardiac arrest to neurological outcomes and mortality [49].

#### 2.1.3. Percutaneous Coronary Intervention (PCI)

Percutaneous coronary intervention (PCI) involves the insertion of an angioplasty balloon catheter guided over a wire across an area of atherosclerotic plaque. The angioplasty balloon is dilated to open an area of stenosis, arterial narrowing or occlusion, to restore blood flow. Balloon angioplasty restores normal blood flow to areas of myocardium that are ischemic, with reduced blood and oxygen supply. In most cases a stent is delivered on the angioplasty balloon to the site of vessel occlusion. PCI in ST-elevation myocardial infarction (STEMI) is the gold standard of care and has revolutionized the management of this life-threatening condition, offering rapid and effective restoration of blood flow to the heart and improving survival.

However, in contrast to the pro-inflammatory role of uPA in the formation of atherosclerotic plaque, where invasive macrophages drive rupture and thrombosis, uPA, or urokinase (UK), was in fact initially developed as a thrombolytic agent that was used to dissolve coronary clots during heart attacks. UK (uPA) was used as a thrombolytic to dissolve clots before coronary stent implants for primary PCI were available or tested for the treatment of STEMI. In 1995, Mitchell and colleagues published a study on patients that revealed a decrease in coronary thrombotic stenoses following the local infusion of 150,000 units of UK administered over a 30 min period [50]. In their in vitro studies, they demonstrated decreased thrombus weight (66% decrease) compared to the control groups with no treatment (25% decrease) when administering UK by direct infusion with a dispatch catheter. In vivo studies further validated the capacity for UK to localize to sites of thrombosis for up to 5 h. Ultimately, the intracoronary infusion of UK resulted in a reduction in the amount of drug required and a more localized treatment. A study involving 345 patients presenting with an acute ST-elevation myocardial infarction (STEMI) caused by a high-grade thrombus were given intracoronary UK prior to stenting. Researchers found more complete ST-segment resolution with the UK group compared to saline treatment [51]. Additionally, the peak CK-MB levels were lower in patients treated with UK. The incidence of bleeding with UK was not significant when compared to untreated controls. A meta-analysis of five randomized, controlled studies with a total of 761 subjects found that UK intracoronary injection increased perfusion to the myocardium, improved cardiac function, and reduced infarction size [52].

In the pursuit of enhancing thrombolytic therapy, researchers have explored innovative strategies, one of which involves the utilization of urokinase-coated nanoparticles (UK-NPs). According to Jin and colleagues, UK-NPs sustained uPA activity, exhibiting an enhanced thrombolytic function over non-conjugated uPA in rabbit models [53]. Additional studies have displayed similar outcomes, affirming the utilization of nanoparticle-conjugated urokinase as a potential combined therapy for thrombolysis [53]. A recombinant form of pro-uPA (known as Prolyse^®^; developed by Abbott Laboratories) has been developed for managing thromboembolic disorders. However, this medication has not yet received approval from the FDA [54]. These studies examining the role of UK as a fibrinolytic further identify the complexity of the uPA/uPAR complex as both a therapeutic thrombolytic as well as a potential pro-inflammatory mediator and marker for disease progression.

Thus again, the complex interplay between cell surface and plasma uPAR (suPAR) levels and the progression of coronary artery disease (CAD) with inflammation and plaque rupture illustrates the multiple roles of uPA and uPAR in vascular disease. The use of uPA (UK) as a beneficial therapeutic, clot-dissolving thrombolytic appears at variance with the increased inflammation of atheroma in vascular disease associated with increased circulating uPAR and the potential to target the uPA/uPAR complex as a new treatment paradigm. UK is used as a thrombolytic treatment in acute MI. However, elevation of plasma suPAR concentration is associated with increased severity of CAD, and is now considered a predictor of future adverse cardiac events [55]. uPAR, thus, has a pivotal role in atherosclerosis progression and plaque rupture, guiding atherogenic cellular invasion and proliferation. A rise in uPAR levels mirrors the release of suPAR from inflammatory cells present at the site of atherosclerosis [55]. The regulation of this balance between the pro-inflammatory functions of uPA/uPAR in driving cardiovascular disease (CVD) and the therapeutic benefit of uPA as a fibrinolytic remains incompletely defined.

#### 2.1.4. Viral Myocarditis

Viral infection, and specifically Coxsackievirus-B3 (CVB3) infection, can lead to myocardial inflammation and damage, termed viral myocarditis. CVB3 myocarditis is characterized by infiltration of inflammatory cells into the myocardium, resulting in cardiomyocyte damage and cardiac dysfunction (heart failure). With viral infection there can be direct damage to the myocardium by viral infection as well as damage induced by the body’s response, a pro-inflammatory response. Both can increase cardiac damage, remodeling, and cardiac dysfunction. There is thus a complex interplay between viral infections, the host inflammatory response, and cardiac function. In mouse models, uPA-deficient mice (uPA-genetic-knockout mice) were tested and found to be protected against remodeling of cardiac tissue after viral infection, demonstrating reduced heart dilatation and failure after CVB3 viral myocarditis [58,129].

#### 2.1.5. Congenital Heart Block (CHB)

Maternal autoantibodies and the uPA/uPAR system are reported to affect fetal cardiac health. Autoantibodies can cross the placental barrier, entering fetal circulation and triggering events that disrupt normal cardiac development, manifesting as structural abnormalities and affecting the cardiac conduction system. CHB is a rare but serious condition, potentially necessitating lifelong medical interventions [130]. Due to circulating maternal anti-SSA/Ro and anti-SSB/La, the fetal heart can become susceptible to fibrosis due to a loss of healthy cardiomyocytes rather than the removal of dead or apoptotic cardiomyocytes. This fibrosis can cause progressive heart block [130,131]. In mouse studies, Park et al. demonstrated that uPAR has a role in phagocytic clearing of damaged cells [59]. Flow cytometry of cell cultures highlighted the localization of uPAR on the surface of apoptotic cardiomyocytes. This suggested an interplay between anti-Ro binding and increased uPAR gene expression. As a result, there is increased scar and collagen deposition causing heart block [60]. These observations highlight the importance of continued research into the role of uPA and uPAR in CHB, with the potential to improve diagnostic and therapeutic strategies, and ultimately improve the care and outcomes of affected infants and their families.

#### 2.1.6. Chronic Heart Failure

The incidence of heart failure has increased, and heart failure is associated with high mortality. An accurate prognostic indicator for the detection of heart failure would be of benefit. As discussed above, suPAR is the cleavage product of membrane-bound uPAR induced by circulating immune cells, and levels of suPAR mirror increased inflammatory responses. In prior work on chronic heart failure, suPAR concentration was strongly correlated to increased mortality in patients [56]. This strong correlation between suPAR levels and mortality highlights the role of uPAR and immune activation as a method for identifying the risk of heart failure and progression. By monitoring suPAR levels, clinicians gain a powerful tool to gauge the intensity of the inflammatory response, which is intrinsically linked to the progression of heart failure.

#### 2.1.7. Cardiovascular Risk Stratification

In a 10-year prospective cohort study of 1951 apparently healthy patients, it was discovered that suPAR, together with markers already in clinical use, e.g., hs-CRP and NT-proBNP, predicted cardiovascular death. In this study, suPAR was suggested as a prognostic biomarker for cardiovascular death in otherwise healthy patients [57].

#### 2.1.8. Acute Ischemic Stroke

In a large population-based cohort study of 6103 subjects, among the key observations, a compelling correlation emerged, drawing a direct link between heightened uPAR levels and the presence of carotid plaques in individuals. This finding is of great clinical relevance, as carotid plaques serve as harbingers of potential local occlusive and embolic atherosclerotic vascular thromboses, with risk of stroke. This study also went on to uncover a heightened risk of ischemic stroke among patients who exhibited increased uPAR levels [61,132]. This study not only adds to the growing body of evidence linking uPAR to vascular pathology, but also underscores its clinical significance as a potential tool for improved risk assessment and personalized management in the realm of cerebrovascular health.

In studies exploring interventions to modulate suPAR levels, it has been observed that individuals on extended courses of beta-blocker treatment exhibit decreased suPAR levels, as demonstrated in carotid endarterectomy biopsy samples [63]. This intriguing link prompts further investigation into the precise mechanisms by which beta-blockers influence suPAR, with the potential to unveil novel therapeutic strategies and diagnostic applications [63].

An acute ischemic stroke, termed a cerebrovascular accident (CVA), is a medical emergency characterized by acute occlusion of cerebral blood vessels causing disabling neurological deficits. Rapid intervention is necessary to restore blood flow and minimize long-term brain damage. Early thrombolytic treatment or mechanical interventions to dissolve or remove blood clots, when given early after the onset of the CVA, improves outcomes, but with an associated increased risk of hemorrhage. uPA (UK) has been employed in the treatment of acute ischemic stroke, as it can help dissolve blood clots in the cerebral vasculature, potentially improving neurological outcomes [61].

The PROACT trial (Prolyse in Acute Cerebral Thromboembolism) is a pivotal trial in the treatment of acute thromboembolic strokes. This phase II randomized human trial indicated that the combination of pro-urokinase and heparin resulted in an improved recanalization efficacy for individuals with acute thromboembolic strokes [62]. This breakthrough not only suggests a potential therapeutic avenue but also underscores the critical importance of addressing the intricate mechanisms underlying thromboembolic strokes. The improved recanalization efficacy observed in the PROACT trial promises to enhance patient recovery and reduce neurological sequelae [62]. Additional clinical trials utilizing uPA as a thrombolytic agent have demonstrated positive results, consistently demonstrating enhanced recanalization, improved functional outcomes, and rates of cerebral hemorrhage that mirror those seen with recombinant tissue plasminogen activator (rtPA) [61].

Here, again the role of suPAR as a diagnostic indicator for unstable disease demonstrates a correlation and a potential role for uPA and suPAR in disease progression. A pro-inflammatory role for uPAR in cerebrovascular disease appears superficially to be contradictory to the use of uPA as a thrombolytic. However, these studies indicate correlations, they do not prove cause. With these various studies, the question arises as to whether the local inflammatory responses driven by the uPA/uPAR complex represents an early disease stage or is limited to local events in the vasculature or may represent a secondary or rebound response. In summary, the use of higher doses of uPA (UK) to dissolve clots has therapeutic benefit, but may also represent a non-physiological response to the high dose of uPA, whereas local uPAR activity may represent a physiological, pathological tissue response. Analyses of local and individual cellular responses to uPA/uPAR activation will provide new insights into the physiological roles of uPA/uPAR in inflammatory responses in contrast to the use of uPA (UK) as a thrombolytic.

#### 2.1.9. Angiogenesis

Angiogenesis is the formation or growth of new blood vessels from existing ones. Angiogenesis plays a crucial role in normal physiological and also pathological conditions, such as wound healing, tissue repair, and tumor growth, and is tightly regulated by a complex interplay of pro-angiogenic and anti-angiogenic factors [133]. uPA is directly involved in the release of pro-angiogenesis growth factors such as FGF-2 and VEGF, playing a key role in endothelial cell proliferation [134].

uPA binding to uPAR on the surface of vascular smooth muscles induces an intricate intracellular signaling cascade which ultimately results in increased migration of cells [64]. Downstream of uPA/uPAR binding, the activation of the JAK-STAT kinase pathway results in platelet-derived growth factor receptor-beta (PDGFR-β) internalization via the LR11 receptor [65].

Studies exploring angiogenesis in human umbilical vein endothelial cells have also demonstrated impaired VEGF-mediated signaling with the knockdown of uPAR. This is further supported by studies in uPAR-deficient mice, where incomplete angiogenesis was observed [66]. Angiogenesis and the growth of blood vessels is necessary to support tumor growth, supplying blood, oxygen, and nutrients to the tumor. In the next sections, we will discuss studies linking cancer progression to uPAR (Section 2.2) and subsequently discuss the role of uPAR in inflammation and disease (Section 2.3). One study investigating endothelial colony-forming cells with properties of endothelial progenitor cells found that whole-uPAR localization in caveolae is required for further angiogenic proliferation [135].

### 2.2. Cancer

The level of suPAR in the blood has been extensively studied as a marker for cancer progression, metastases, and abnormal immune responses to cancer in patients (Table 1). Aberrant immune responses are now understood to drive tumor growth and invasion, both in local tumor- and stromal-associated mononuclear cells and in systemic immune cell responses. Increases in suPAR have been detected in patients with metastatic cancer. Based on these observed increases in uPAR with metastatic cancer, new approaches that target uPAR have been developed using small molecules, selective anti-uPAR antibodies, uPA-mediated toxin activity, uPAR-targeting nanoparticles designed to transport chemotherapeutic agents for local delivery to the tumor, and one virus-derived immune-modulating serpin. Several studies have reported reduced cancer invasion and metastasis with these new uPAR-targeting reagents. Here, we describe a variety of cancers for which suPAR and uPAR have been identified as markers for cancer progression and metastasis and as new therapeutic approaches.

#### 2.2.1. Cancer Metastasis

Recent studies have indicated a pivotal role for uPAR in malignant tumor invasion and metastasis [67]. High uPAR expression is consistently observed in solid tumors. uPAR is intricately involved in extracellular matrix degradation, tumor angiogenesis, cell proliferation, and apoptosis, as well as multidrug resistance (MDR) in cancer cells (Figure 2 and Figure 3). Here, we discuss specific malignancies in which associated changes in uPAR expression have been identified in breast, lung, prostate, head and neck, leukemia, ovarian, and pancreatic cancers (Table 1).

#### 2.2.2. Breast Cancer

Breast cancer is the most common cancer among females and the incidence is estimated at 2.3 million new cases globally each year [68,136]. Recent studies have demonstrated an increase in the expression of uPA, uPAR, and plasminogen activator inhibitor type-1 (PAI-1) in breast cancer and bony metastases. Studies have analyzed tissue samples from various breast carcinomas, metastases, and normal breasts, identifying associated changes in uPAR. The majority of the tumors analyzed had moderate uPA mRNA levels and variable uPAR and PAI-1 mRNA levels, primarily localized in epithelial tumor cells. Malignant tumors exhibited significantly increased uPAR mRNA expression as well as slightly increased uPA and PAI-1 mRNA expression on comparison to benign breast tissue [68,69,71].

A specific subset of breast carcinomas expressing human epidermal growth factor receptor type 2 (HER2-positive) display high uPAR expression. HER2-positive breast cancer is known for an aggressive and metastatic nature, although treatment with targeted therapies is available and effective with early treatment. A microarray analysis of advanced breast cancer lesions identified interactions between HER2 and uPAR, along with a wide array of downstream molecules. The HER2-positive/uPAR-positive subtype demonstrates increased expression of transcriptional factors, contributing to the aggressive nature of this tumor. This overexpression of uPAR may suggest a reason for resistance to treatment in HER2-positive cancers and the potential for uPAR to provide a new therapeutic target [68,69,71]. Antibody drug constructs that target uPAR are in development as potential treatments for triple-negative breast cancer [71].

#### 2.2.3. Lung Cancer

Bronchogenic lung carcinoma is the leading cause of cancer-related deaths in the United States. Smoking is the primary cause of lung malignancies, but it is further exacerbated by exposure to environmental factors such as asbestos and polycyclic aromatic hydrocarbons [137]. Two large-cell lung carcinoma strains in humans, one highly metastatic (strain 95D) and one less metastatic (strain 95C), were evaluated for their in vitro and in vivo invasive and metastatic potentials. Strain 95D exhibited greater invasiveness than strain 95C. The expression levels of uPA and tPA, uPAR, and PAI-1 and PAI-2 were assessed by RT-PCR and immunohistochemical staining in the high- and low-metastatic strains [72]. The high-metastatic strain 95D displayed elevated uPA and uPAR expression and reduced tPA and PAI-2 levels compared to the low-metastatic strain 95C. PAI-1 expression was similar in both strains. Monoclonal antibodies targeting uPAR effectively decreased the invasive potential of strain 95D cells in vitro. This suggests that uPAR plays a significant role in the invasiveness of these lung carcinoma strains [72].

Small-cell lung carcinoma (SCLC) is a subset of highly proliferative lung carcinoma that presents with rapid growth, early metastasis, and poor prognosis [73]. Studies have also investigated the role of uPAR in six different SCLC cell lines. The findings revealed that a subpopulation of cells in these SCLC lines expressed uPAR. These uPAR-positive cells exhibited resistance to multiple drugs, with high clonogenic (survival) activity, and also co-expressed CD44 and MDR1, which are markers of potential cancer stem cells. This work suggests that uPAR-positive cells represent a functionally important subset of chemotherapy-resistant cancer cells in SCLC and could be valuable targets for more effective therapeutic interventions [74].

The overexpression of uPAR in a range of lung cancer subtypes further suggests a role for uPAR as a suitable target for chemotherapy. Peptide sequences of the amino-terminal domain of uPA have recently been proposed as an efficient method to target uPAR in lung cancer cells. Investigation of the uPAR-targeting U11 peptide conjugated with a pH-sensitive doxorubicin and curcumin combination has demonstrated a synergistic anti-tumor effect on cultured non-small-cell lung cancer cells in vitro and in vivo in a mouse model [75].

#### 2.2.4. Prostate Cancer

Prostate cancer (PCa) is a heterogeneous disease involving genetic, environmental, and social influences, commonly affecting men aged 45 to 60 [138]. A recent analysis of uPA, uPAR, and PAI-1 in patients undergoing radical prostatectomy for cancer, identified increased levels of these proteins. An immunohistochemical analysis of tissue samples from 3121 patients revealed overexpression of uPA, uPAR, and PAI-1 in varying proportions [70].

Overexpression of uPAR was associated with aggressive PCa. All three markers were linked to an increased risk of cancer recurrence as detected on biochemical assay. The likelihood of recurrence increased with a higher number of overexpressed markers. A decision curve analysis indicated that the inclusion of data for uPA, uPAR, and PAI-1 improves clinical decision-making when compared to standard clinical–pathological features [70]. A follow-up analysis using tissue microarrays and immunohistochemistry found that overexpression of uPAR was present in more than half of primary PCa tissues and over 90% of lymph node metastases, but not in normal or benign tissues. The overexpression of uPAR was further associated with higher Gleason scores, indicating a link between their expression and tumor differentiation, even in patients with favorable pathological characteristics [76].

The high prevalence of uPAR in PCa suggests its potential as an effective therapeutic intervention. Noscapine, an anticancer compound, has shown promise in inhibiting tumor growth in various types of cancers, but its effectiveness has been limited by bioavailability. With recent developments in nanoscale delivery systems, researchers have utilized the human-type amino-terminal fragment of uPA as a natural ligand for uPAR to improve drug targeting and availability [77]. uPAR-targeted nanoparticles significantly enhanced the intracellular accumulation of noscapine in PCa, leading to a more potent inhibitory effect compared to free noscapine [77]. Further research identified the effects of uPAR knockdown using small interfering RNA (siRNA) or combined treatment with microRNA (miRNA) and siRNA. Systemic treatment with tumor suppressor gene MIR143 in polymeric nanoparticles inhibits tumor growth in mice with subcutaneous PC-3 tumor xenografts. The nanoparticle-mediated delivery of MIR143 significantly downregulated uPAR protein levels without affecting mRNA levels, indicating translational inhibition [78].

#### 2.2.5. Head and Neck Cancer

Head and neck cancers are notorious for their late discovery, as most patients initially present with lymph node metastases [139]. In comparison to more common cancers, head and neck squamous cell carcinoma (HNSCC) has a poorly described mechanism for invasion and metastasis. Therefore, identification of biological markers may enable earlier diagnosis and the potential for new treatment targets. The plasminogen activator system, especially uPA and uPAR, plays a significant role in head and neck squamous cell carcinoma (HNSCC). The overexpression of uPAR and PAI-1 (SERPINE1 gene) is reported to contribute to increased tumor cell migration, invasion, and metastasis, ultimately resulting in a poor prognosis. Both uPAR and PAI-1 are associated with the induction of epithelial-to-mesenchymal transition, the acquisition of stem cell properties, and resistance to anticancer agents [79].

#### 2.2.6. Leukemia

Leukemias are hematologic cancers with a wide prevalence in both pediatric and adult patients. The major sub-classifications of leukemia encompass chronic lymphocytic leukemia (CLL), acute myeloid leukemia (AML), chronic myeloid leukemia (CML), and acute lymphoblastic leukemia (ALL) [140]. While the primary focus of uPAR analysis has been on the effect of the translated protein, recent research has indicated that transfection of uPAR 3′UTR in AML tumor cells affects the expression of pro-tumoral factors, cell adhesion, and migration. These findings demonstrated that uPAR 3′UTR-recruited microRNAs can modulate multiple intracellular functions by targeting various transcripts. Additionally, variants of uPAR transcripts with 3′UTR regions are detected in U937 leukemia cells, with higher uPAR expression [83]. These findings suggest that uPAR mRNA variants play an important role in the progression of AML.

#### 2.2.7. Ovarian Cancer

Ovarian cancer is a leading cause of death among women with gynecologic cancer. Similar to HNSCC, as discussed previously, ovarian cancer is typically diagnosed at a later stage, leading to worse outcomes [139,141]. A major driver in ovarian cancer (OC) is lysophosphatidic acid (LPA), which stimulates both cellular migration and proliferation. A recent analysis of uPAR expression in ovarian epithelial cancer cells stimulated with LPA revealed an increase in uPAR aggregation and uPA binding [80].

More recently, the Kazal-type serine protease inhibitor-13 (SPINK13) gene was reported as associated with decreased mortality in OC patients. Analysis of the SPINK13 molecular pathway identified its role in inhibiting the expression of uPA, further emphasizing the significant role of uPA in OC [81]. Similarly, subtypes of ovarian cancer, such as leptin-induced OC cell invasion, rely heavily on uPA.

#### 2.2.8. Pancreatic Cancer

uPAR expression has been reported to be highly expressed in pancreatic cancer, more so than many other cancers. Treatment of a xenograft model of pancreatic cancer cells in SCID mice with the virus-derived immune-modulating serpin, Serp-1, demonstrated significant reduction in tumor growth along with altered stromal and regulatory T cell immune responses [82]. Serp-1 requires uPAR for normal immune-modulating functions (Table 2).

### 2.3. Inflammation

#### 2.3.1. Chronic Inflammation

Chronic inflammation (CI) is characterized by persistent, low-grade innate immune-cell activation that significantly impacts an individual’s quality of life and contributes to the development of disease. Currently, CI lacks well-defined diagnostic criteria, often relying on markers associated with acute inflammation, despite its long-lasting, chronic nature, spanning years [159]. CI is associated with chronic diseases such as DM, CVD, arthritis, and renal disease, and even long-term infections such as tuberculosis. The transition to CI inflammation typically occurs when the body struggles to repair and resolve an initial acute inflammatory response and associated damage leading to ongoing excess systemic immune-cell activation and tissue damage. Recent research has investigated the response of immune cells under inflammatory stimuli and has revealed an increase in suPAR (Table 1). Blood suPAR demonstrates a strong correlation with inflammation as well as a strong association with increased circulating immune cells [55]. The role of suPAR is further supported by the fact that suPAR shares common risk factors for age-related diseases as mentioned in this review, including atherosclerotic coronary and carotid artery disease and COPD [31,55].

Interestingly, in contrast to CRP, suPAR plasma levels remain unaffected by circadian fluctuations, with relatively steady expression even during periods of acute stress [84]. For instance, serial measurements demonstrated a mere 15% average increase in suPAR levels, in stark contrast to the 365% rise in hs-CRP levels in the setting of an MI. Overall, suPAR was reported to be a better prognostic indicator for adverse outcomes than hs-CRP [55].

Macrophage invasion of atherosclerotic plaque in coronary arteries leads to plaque rupture and acute thrombosis in MI, as noted above in Section 2.1. Unstable coronary plaque is, thus, an example of an inflammatory disorder and the uPA/uPAR complex is associated with activation of matrix metalloproteinases (MMPs) that allow for tissue breakdown and inflammatory cell invasion into plaque and into the arterial wall. suPAR appears to outperform hs-CRP in terms of prognosis for hospital mortality for individuals with coronary artery disease and has been established as an independent predictor for future adverse cardiac events [55,56,57,84]. The utilization of suPAR as a diagnostic marker for CI represents an exciting avenue of research.

#### 2.3.2. Rheumatoid Arthritis

Rheumatoid arthritis (RA) is a complex autoimmune disease associated with various genetic and environmental factors. RA typically begins unilaterally in a peripheral joint and subsequently progresses to proximal joints, resulting in cartilage loss and bone erosion [160]. RA often has a delay in initial diagnosis. Extensive analysis has revealed that major histocompatibility complex, class II, DR beta 1 (HLA-DRB1) carries a segment of five conserved amino acids in the hypervariable regions strongly associated with RA development.

Earlier investigations on the assessment of circulating suPAR in RA demonstrated elevated suPAR levels in RA patients compared to their healthy counterparts [85,86,161]. These suPAR levels exhibited a direct correlation with the number of inflamed joints, even among patients with limited disease activity [85]. In a recent study involving 252 patients from a Swedish prospective observational cohort with early RA, serum suPAR was evaluated using an enzymatic immunoassay at disease onset, as well as after 3 and 36 months [160]. In contrast, tPA is generally downregulated in RA. Based on current research, suPAR detection has the potential to serve as an adjunctive tool for the screening and monitoring of disease progression in patients with RA [85,86,160,161,162].

### 2.4. Critical Care

#### 2.4.1. SIRS—Systemic Inflammatory Response Syndrome

SIRS (systemic inflammatory response syndrome) is associated with high morbidity and mortality, with or without an antecedent infectious process (sepsis). A study of 132 patients investigated the utility of suPAR as a biomarker for SIRS, and the capacity for suPAR to differentiate SIRS from bacteremia in comparison with other biomarkers. suPAR was found to have an AUC (area under receiving operator curve) of 0.726 in differentiating SIRS from sepsis; an acceptable biomarker for differentiation. When combined with other current clinically used biomarkers, procalcitonin and IL-6, the AUC rose to 0.804, which is considered a good classification level for differentiating SIRS from sepsis. Interestingly, initial suPAR concentrations were significantly higher in patients who later died within 28 days, supporting its utility as a prognosticator for overall inpatient mortality upon initial presentation [163].

#### 2.4.2. Sepsis

Sepsis is defined as a systemic inflammatory response syndrome (SIRS) in response to a confirmed infectious process such as a bacteremia or viremia. Sepsis is, thus, considered a form of SIRS but with an identified source of infection as the culprit. Sepsis is a serious condition produced by infections with bacteria and also viruses, fungi, and parasites. In sepsis, there is an extreme immune response causing systemic symptoms of fever, tachycardia, tachypnea, and shock, with associated high mortality. Sepsis is a major cause of morbidity and mortality in newborns. A meta-analysis conducted in 1959 patients over six studies concluded that suPAR had superior specificity for differentiating neonatal sepsis, by clinical definitions, from non-septic neonates when compared to the commonly used procalcitonin and CRP levels. Furthermore, the diagnostic odds ratio was 117, indicating high efficiency and precision. The positive likelihood ratio was 14, indicating that the suPAR level is 14 times higher in neonates with sepsis than those without. Finally, the negative likelihood ratio was low at 0.12. This means that if the suPAR level is negative, the probability of neonatal sepsis is 12%, allowing confidence in ruling neonatal sepsis. Interestingly, in neonatal patients, and in concordance with the adult patients, the measured levels of suPAR on the first day helped in predicting mortality in late-onset neonatal sepsis [88,89,90]. Finally, suPAR is also useful as a prognostic biomarker in adult sepsis [91].

### 2.5. Wound Healing and Tissue Repair

Tissue repair is a complex process that entails the orchestrated activation of diverse intracellular mechanisms that facilitate cellular migration, proliferation, and differentiation [164,165]. These intricate pathways are subject to a very rigorous regulation, as their dysregulation can predispose individuals to a wide spectrum of infections and malignancies. The skin’s dermal layers have both normal structural cells but also intrinsic or local tissue immune cells that can act as a primary response to tissue damage, similar to the central nervous system. As noted, uPAR plays a pivotal role in cellular migration, with increased prevalence in injured tissues (Table 1).

uPAR has been identified as playing a major role in both epidermal and dermal wound healing environments. Recent investigation of overexpressed demogelin 2, an important regulator of cell survival and proliferation, identified an increased release in uPAR expression [166]. Leveraging the increased activity of the uPAR signaling pathway in the context of wound healing holds promise for the identification of pharmacological interventions that can expedite tissue repair. Recent investigations have unveiled the significance of spermidine (SPD), an abundant natural polyamine essential for endothelial cell proliferation and angiogenesis, that strongly promotes the activation of uPAR when administered both systemically and topically [167,168]. Initial in vitro scratch wound assay studies in tissue cultures provided evidence of spermidine/uPA/uPAR wound healing-promoting properties.

Subsequent in vivo analyses were conducted in a murine model of skin-wound repair to determine the effectiveness of topical SPD treatment. Notably, mice treated with SPD exhibited substantial enhancements in the healing. Examination of the wounds not only revealed heightened activity within the uPA/uPAR signaling pathway but also demonstrated increased expression of interleukin-6 (IL-6) and tumor necrosis factor (TNF), two pivotal proteins associated with both inflammation with potential damage as well as with tissue regeneration [168]. Further research is needed to better understand the role of uPA/uPAR in wound healing as well as identify novel promoters of this pathway.

### 2.6. Neurology

uPAR has also been reported to have a role in neuronal damage and repair (Table 1). The CNS also has an intrinsic immune system that includes glial and astrocyte cellular actions. Through immunohistochemical analysis of necrotic brain lesions and focal cerebral infarcts, continuous uPAR expression was identified up to four days post-injury, with a peak at the 12 h mark [111].

In a study that investigated the expression of uPAR in the murine central nervous system (CNS) during inflammatory responses, suPAR was identified in the CSF in various diseases or pathologies with increases in the CSF in patients with HIV dementia, a chronic neuroinflammatory disease [24,111,112]. There is significant upregulation of uPAR at both mRNA and protein levels in microglial cells during acute intracerebral lipopolysaccharide (LPS) exposure. Expression of uPAR in neuronal cell lines with subsequent binding to vitronectin, a protein located predominantly in the ECM, have shown to cause extensive changes in cellular morphology and actin cytoskeleton formation [24].

uPA expression and activity are prominently increased during chronic neurodegeneration, suggesting potential proteolysis-independent roles for uPAR in acute disease. In both kainate and LPS challenges, uPAR was significantly upregulated, whereas uPA was not altered. Further investigations have found a correlation between suPAR levels in the CSF with the presence of any form of CNS inflammation, e.g., not with specific pathological conditions, as highlighted by the diffuse staining found on histology [111].

Various kinds of inflammation are predicted to prompt release of suPAR from microglial cells membranes. For instance, the CSF of patients with human immunodeficiency virus (HIV)-associated dementia had elevated levels of suPAR, yet specific demyelinating pathologies such as multiple sclerosis or Guillian–Barre syndrome did not show altered suPAR levels [112]. The determination of resident versus reactive resident microglia and invasive systemic inflammatory macrophage and lymphocytes that express suPAR as a response to neuroinflammation remains under investigation.

PAI-I is upregulated in a variety of neurodegenerative states including Parkinson’s disease and Alzheimer’s disease (AD). Plasmin cleaves and degrades α-synuclein, and α-synuclein upregulates PAI-1. It has been proposed that an excess of PAI-1 in the brains of PD patients prevents plasmin-induced clearance of α-synuclein aggregates, which is corroborated by the fact that elevated PAI-1 levels are associated with worse outcomes for PD patients. Finally, PAI-1 is downregulated, with associated upregulation of tPA and uPA, during exercise in patients with AD [169]. As suPAR is more bioavailable than uPAR, it is reasonable to hypothesize that suPAR may contribute to plasmin-associated alpha-synuclein disease states, providing a biomarker.

In the assessment of stroke risk, suPAR levels were elevated in patients with symptomatic carotid artery plaques in a study on 162 patients, supporting the use of suPAR as a biomarker for unstable carotid artery atherosclerosis at risk of thrombotic occlusion [114]. Additionally, suPAR was found to be superior to CRP in specificity for discrimination between vertebral osteomyelitis and other neurodegenerative spinal diseases in a 36-person study [114].

### 2.7. Infectious Disease

uPAR has been associated with infectious disease, specifically, HIV, SARS-CoV-2, and tuberculosis (Table 2).

#### 2.7.1. HIV

In addition to the studies examining bacterial sepsis in critical care units, increased uPAR expression has also been detected in other infections. Speth and colleagues were the first to identify increased levels of uPAR expression on the surface of lymphocytes and monocytes with HIV infection [92]. Based on these findings, another group of researchers found a strong correlation between prognosis of HIV-1 and blood suPAR levels, similar to the prognostic value of the CD4+ count and viral load, and thus, survivability [93].

#### 2.7.2. COVID-19 Pneumonia

SARS-CoV-2 infection is the coronavirus infection that caused the recent coronavirus pandemic (COVID-19). CoV-2 causes acute lung injury and acute respiratory distress syndrome, stemming from severe inflammatory pneumonia and associated immune and thrombotic vascular damage. It has been proposed that dysregulation in the uPA/uPAR system may induce some of the CoV-mediated inflammatory damage. suPAR levels have been investigated in patients with COVID-19, with suPAR levels reported as a marker for increased risk for intensive care admission. The Ly-6/uPAR family of receptors have been proposed as sites activated by the SARS-CoV-2 S protein, increasing cellular infection by the virus. uPA/uPAR may also provide a potential therapeutic target for treating the dysregulated immune and thrombotic responses in COVID-19 [170,171].

The SAVE-MORE (suPAR-Guided Anakinra Treatment for Validation of the Risk and Early Management of Severe Respiratory Failure by COVID-19) trial examined 606 patients randomized to a phase 3 controlled trial. Initially, 1060 patients were enrolled; however, those with suPAR levels less than 6 ng/mL were excluded. Based upon suPAR stratification, patients treated with anakinra experienced a survival benefit with a 3.9% 28-day mortality, markedly increased when compared to the 8.7% mortality at 28 days in control subjects. Thus, suPAR has shown efficacy in the improvement of patient selection criteria as a useful biomarker in patient stratification to assist patients with COVID-19 infections treated with anakinra [116].

Of those who were excluded due to not reaching threshold suPAR levels (i.e., <6 ng/mL), a post hoc analysis revealed that only 2.9% of those patients progressed to respiratory failure or death, yet again highlighting the importance of suPAR as an early inpatient biomarker and major predictor of impending intrahospital morbidity and mortality [116].

In a mouse-adapted SARS-CoV-2 (MA30) model, PEGSerp-1 treatment reduced weight loss, clinical symptoms, inflammation, and damage to lung and vascular tissue (Table 2) [151]. Of particular interest, tissue uPAR detected by IHC was reduced by PEGSerp-1, but PCR (polymerase chain reaction) analysis indicated increased uPAR gene expression.

#### 2.7.3. Tuberculosis

In a community study, individuals with an active infection of *Mycobacterium tuberculosis* had significantly increased levels of suPAR using ELISA detection assays [94]. After an 8-month treatment period, suPAR levels decreased to levels of tuberculosis (TB)-negative subjects. Therefore, suPAR may provide a biomarker for treatment efficacy in TB.

### 2.8. Nephrology

#### 2.8.1. Acute Kidney Injury (AKI)

suPAR was elevated in 3827 patients undergoing coronary angiography, 250 patients undergoing cardiac surgery, and upon admission in 692 critically ill ICU patients. Those within the upper quartile for suPAR levels had an increased risk of acute kidney injury (AKI) and death at 90 days across all three cohorts. Mice studies performed by the investigators revealed that mice overexpressing suPAR and mice given suPAR when given contrast for imaging had greater pathologic evidence of AKI, energetic demand, and mitochondrial superoxide generation than wildtype mice. Pre-treatment in these mice with a monoclonal uPAR antibody reduced AKI in mice overexpressing uPAR, normalizing oxidative stress [102].

#### 2.8.2. Focal Segmental Glomerulosclerosis

Focal segmental glomerulosclerosis (FSGS) causes roughly 20% of all glomerular disease and represents a major cause of end-stage renal disease (ESRD) requiring hemodialysis. Two-thirds of all patients with FSGS will have elevated suPAR levels (using 3.0 as the cutoff), which is significantly higher than all other glomerular diseases. Additionally, higher suPAR levels were predictive for FSGS recurrence following kidney transplantation. 54 patients with FSGS were found to have elevated suPAR levels when compared to patients with glomerulonephritis and healthy control subjects. The optimal cut-off value for diagnosis of FSGS using suPAR was found to be 4.644 ng/mL, with a sensitivity and specificity of 0.91 and AUC of 0.946, providing an excellent diagnostic test for FSGS. The authors supported suPAR as a biomarker for a variety of purposes, some of which will require more randomized controlled trials for individualized immunosuppressive therapy in FSGS recurrence post-transplant, ESRD prediction, and diagnosis of other glomerular diseases when kidney biopsy is not possible. SuPAR may be most useful when combined with other established biomarkers, such as anti-PLA2R antibodies [103].

### 2.9. Gastroenterology

uPAR has also been examined as a marker for severe gastrointestinal disorders such as pancreatitis, hepatitis, and inflammatory bowel disease (Table 1).

#### 2.9.1. Acute Pancreatitis

Acute pancreatitis is inflammation of the pancreas leading to increased capillary permeability. Increased capillary leak is considered an antecedent to non-septic SIRS with potential to cause organ failure and multi-organ distress syndrome. The severity of pancreatitis ranges from mild pancreatitis, causing local complications without signs of organ failure; to moderate, showing signs of organ failure recovering within 48 h; and finally, to severe, leading to organ failure persisting past 48 h [172]. Given its potential for morbidity and mortality, predicting the type of pancreatitis is important for planning treatment modalities.

Researchers conducted a study on 225 hospitalized patients with acute pancreatitis. Of those, 75 had severe acute pancreatitis (SAP), 75 had moderate–severe acute pancreatitis (MSAP), and 75 had mild acute pancreatitis (MAP). Another 75 healthy patients served as controls. Serum samples of suPAR were taken 24 h after admission. suPAR could significantly differentiate SAP from healthy controls with high diagnostic accuracy (AUC = 0.920). SAP was statistically significant, differentiating MAP with an AUC of 0.855 and MSAP with an AUC of 0.684. suPAR correlated with clinically used lab values and clinical scores including Ranson’s score and CRP. Finally, suPAR was able to predict inpatient mortality from SAP with an AUC of 0.806. The study concluded that suPAR could be used as a potential biomarker for inflammation, severity, and inpatient mortality in SAP patients [95].

#### 2.9.2. Chronic Hepatitis and Fibrosis Progression

Twelve potential biomarkers were evaluated in twenty-one cirrhotic patients positive for chronic hepatitis C (HCV) genotype I versus twenty-one healthy controls, assessing for progression of liver fibrosis as evidenced by transient elastography. suPAR was found to have an AUC of 0.78, a fair diagnostic biomarker for progression of liver fibrosis in chronic HCV patients [95]. A later study by another research group corroborated these findings. Researchers examined 146 chronic infections in HCV patients in two cohorts. Mean suPAR levels were not elevated in the earlier-stage fibrosis F1 and F2 stages, but were significantly increased in the F3 and F4 stages, when compared to healthy subjects. The AUC in distinguishing F1/F2 from F3/F4 was 0.774, and the AUC in distinguishing non-cirrhotic (F1, F2, and F3) from cirrhotic patients (F4) was 0.791. Serum suPAR levels also strongly correlated with noninvasive clinically used biomarkers of fibrosis in the aspartate transaminase-to-platelets ratio index score (r = 0.52) and transient elastography imaging studies (r = 0.44; all *p* values < 0.0001) [96].

The gold standard for monitoring liver progression is invasive liver biopsy for cirrhosis staging in hepatitis B cirrhosis. To this end, a biomarker is needed to assess liver cirrhosis progression. A total of 76 patients and 21 healthy controls were recruited to assess various biomarkers for liver cirrhosis progression. Clinically used biomarkers including mean platelet volume or aspartate aminotransferase-to-platelet ratio index scores failed to achieve statistical significance. However, suPAR and IL-10 were significantly higher in those patients with severe fibrosis versus mild fibrosis [97]. The finding of elevated suPAR levels was corroborated in another study including 105 cirrhotic patients when compared to 19 liver-healthy controls [98].

#### 2.9.3. Acute Decompensated Liver Failure

No biomarker yet exists which separates the severity of decompensated cirrhosis from bacterial infections common in the patient population. In a single-center study of 162 patients with decompensated liver cirrhosis who underwent diagnostic paracentesis at a tertiary care center in Aachen, Germany, suPAR levels were increased in patients with decompensated cirrhosis and correlated with the severity of liver dysfunction and systemic inflammation, but were not indicative of bacterial infection. Serum suPAR levels > 14.4 ng/mL predicted 28-day mortality in these patients. Ascitic fluid suPAR levels were elevated during episodes of spontaneous bacterial peritonitis, but were not elevated during episodes involving bacterial translocation into the ascitic fluid. The reasoning by the authors, uncovered in in vitro experiments, was that monocytes (and to a lesser extent neutrophils) secreted suPAR following TLR activation, leading to rapid uPAR cleavage and upregulation [99,173].

#### 2.9.4. Inflammatory Bowel Disease

Inflammatory bowel disease (IBD), including ulcerative colitis (UC) and Crohn’s disease, is characterized by recurrent chronic inflammation of the intestinal tract. The significant rise in IBD cases in the 21st century has been strongly associated with the rapid industrialization of countries [114]. Despite ongoing research, the exact cause of IBD remains a mystery, posing challenges to diagnosis and treatment given its multifactorial nature. A recent comprehensive transcriptomic meta-analysis of public IBD datasets has identified uPAR as a potential key factor in IBD. Mouse models of IBD have increases in uPAR expression during epithelial cell breakdown. Pharmacological blockade and knockdown of uPAR in the same mouse model demonstrated a protective effect against cytokine-induced mucosal barrier breakdown [174].

The role of uPAR in IBD suggests a potential novel therapeutic target. The breakdown of the intestinal epithelial barrier is a crucial factor in the development of inflammatory bowel disease (IBD), causing inflammation, damage, and loss of crypt architecture. A potential avenue for innovative IBD therapeutics lies in targeting specific ligand–receptor pairs in IBD mucosa, which may play a role in maintaining the integrity of the intestinal epithelial barrier. The uPA–uPAR complex was presented as a therapeutic target for IBD based upon meta-analysis indicating that there is upregulation of urokinase-type plasminogen receptor–ligand genes in damaged mucosa. Interestingly, the same genes were expressed less during barrier formation. This coordinated upregulation of uPA–uPAR in UC and CD biopsies is hypothesized to be linked to cytokine-induced damage to the epithelial barrier [117].

Human intestinal epithelial cell lines with knockout of uPA and uPAR genes have shown increased protective barrier formation, further indicating the dysregulatory role of uPA–uPAR in IBD. In primary-organoid-derived cell monolayers, researchers were able to demonstrate improved barrier function with small-molecule inhibitors, peptide antagonists, and neutralizing antibodies targeting the uPA–uPAR complex. The uPA protease inhibitor did not show the same significant epithelial barrier protection effect. These findings stress the importance of targeting intracellular receptor signaling to attenuate IBD, while also avoiding potential unwanted side effects of uPA inhibition. This paper also highlights the lesser importance of inhibiting ECM formation, which is modified by uPA-uPAR. To gain enhanced insights into the therapeutic effects in vivo, it will be necessary to use a monoclonal antibody and/or a highly effective small-molecule inhibitor to hinder the interaction between uPA and uPAR [117].

A dextran sulfate sodium (DSS) colitis model was tested in either wildtype or uPAR-knockout mice. Mice lacking uPAR exhibited significant protection, evident in the restoration of colon mucosa architecture, a reduction in inflammatory infiltrate, and improved surface integrity. Notably, in wildtype mice treated with DSS, a disruption of cellular tight junctions was observed, and this effect was markedly diminished in the knockout (KO) mice, indicating a safeguarding of the epithelial barrier in vivo by uPAR deficiency. The improved colon morphology and reduced epithelial damage in uPAR-KO mice indicated a protective role of uPAR inhibition against DSS-induced colitis [117].

Since uPAR is a GPI-anchored receptor, it has the capacity to interact with various membrane and intracellular pathways. Through co-immunoprecipitation, Cheng and colleagues found that uPAR–epidermal growth factor receptor (EGFR) binding is undisturbed following barrier damage. In contrast, the association of uPAR with integrin subunits was lost following the cytokine-induced damage [117].

Repair signals from EGFR were also increased in uPAR-deficient cells during barrier breakdown. The signaling pathway of EGFR plays a crucial role in regulating epithelial functions, including maintenance of cell junctions, cell survival, and secretion of mucin [118]. Since EGFR-uPAR binding is unchanged during IBD pathogenesis, this finding could support the key role of uPAR in suppressing EGFR-modulated repair mechanisms and signaling. Upon cytokine challenge, tight junctions were disrupted with cell death increased, indicating dysfunction in EGFR. In uPAR-knockout cells, an overall increase in cell survivability and decreased junction damage was observed, in addition to enhanced activity of 45 different kinases/phosphoproteins associated with EGFR [118].

### 2.10. Endocrinology

#### 2.10.1. Type I Diabetes

suPAR levels were assessed in 667 patients with type I diabetes (T1DM) versus 51 nondiabetic control patients in a single-center cross-sectional study, stratified into associated complications of diabetes. suPAR was elevated across all cohorts with T1DM. suPAR levels were more elevated in T1DM patients with cardiovascular complications. Patients with cardiovascular disease were observed to have a 2.5 times higher suPAR level, autonomic dysfunction carried a 2.7 times greater suPAR level, 3.8 times elevated for albuminuria, and finally 2.5 times higher for patients determined to have stiff arterial walls [100].

#### 2.10.2. Type II Diabetes

SuPAR levels, as in T1DM, are also elevated in patients with T2DM. One study investigated suPAR as an early biomarker for diabetic nephropathy, potentially providing an earlier marker than the use of the clinically approved microalbuminuria. Researchers investigated baseline suPAR levels with incidental microalbuminuria in a prospective longitudinal cohort study of 258 patients at risk of T2DM. Another cohort was studied looking for association with albuminuria at later stages of T2DM in a cross-sectional cohort with diagnosed T2DM. A higher baseline suPAR level was associated with a higher risk of microalbuminuria in subjects at risk for T2DM for the higher quartile of subjects when compared to the lower quartile of subjects. Patients with new-onset microalbuminuria were found to be at increased risk of prediabetes. suPAR levels were consistently elevated in patients with microalbuminuria in a separate cohort of patients with already-diagnosed T2DM. Finally, elevated baseline suPAR concentrations were independently associated with new-onset microalbuminuria in subjects at risk for T2DM. Thus, suPAR may present earlier than microalbuminuria in patients at risk of T2DM [101].

### 2.11. Pulmonology

Lung disease has also been associated with altered uPAR activity (Table 1).

#### 2.11.1. Asthma

Asthma is responsible for a substantial number of hospital readmissions, and at present no prognostic biomarker exists for asthma patients. To this end, serum suPAR and eosinophils were taken upon admission from 1341 patients admitted with an acute exacerbation of asthma. The 365-day readmission and all-cause mortality were assessed. Patients who were either readmitted or died had higher suPAR levels and decreased eosinophil levels upon admission. Patients in the 4th quartile for suPAR levels or eosinophil counts < 150 cells/uL had the highest odds for readmission or mortality. The investigators for this study stated that the concurrent use of suPAR with eosinophils as a biomarker allows for precision in asthma prognostication for patients at higher risk of adverse events or lower disease control [104].

#### 2.11.2. Ventilator-Associated Pneumonia

Ventilator-associated pneumonia (VAP) remains a challenge for timely diagnosis for appropriate treatment. In an observational, prospective, multicenter cohort study of 24 patients with VAP compared with 19 control patients, suPAR levels were found to be significantly elevated in VAP patients three days before definitive VAP diagnosis, albeit poorly, with an AUC of 0.68. The AUCs on the day of diagnosis and in deceased patients were even higher, at 0.77 and 0.79, respectively; when combined with already clinically used C-reactive protein, procalcitonin, and the Clinical Pulmonary Infection Score, VAP prediction and specificity increased. Thus, on its own, suPAR had a fair diagnostic accuracy, but as an adjunct biomarker suPAR further assisted in diagnosis [106]. Furthermore, in a homogeneous Greecian cohort of 180 patients with concurrent VAP and sepsis, suPAR levels greater than a 10.5 ng/mL cut-off had 80% specificity and 77.6% positive predictive value to discriminate between severe sepsis and sepsis. suPAR levels greater than 12.9 ng/mL had 80% specificity and 76.1% positive predictive value for prognosis of unfavorable outcome. Finally, in these critically ill patients with sepsis and VAP, suPAR was identified as an independent factor associated with unfavorable outcome, as identified using stepwise Cox regression analysis [107].

#### 2.11.3. Community-Acquired Pneumonia

Community-acquired pneumonia (CAP) is the most common infectious disease carrying high mortality. In a study investigating suPAR levels in 75 patients with CAP versus 67 healthy controls, suPAR levels were significantly elevated in patients with CAP, and correlated with the clinically used Pneumonia Severity Index scores. LPS was found to induce suPAR expression in macrophages in a mouse model of CAP. The authors concluded that plasma suPAR levels may provide a biomarker for CAP severity, and may potentially serve as a therapeutic target [105].

#### 2.11.4. Chronic Obstructive Pulmonary Disease (COPD)

Chronic obstructive pulmonary disease (COPD) is a chronic inflammatory disease causing hospital readmission and mortality through repeated exacerbation. Commonly used acute-phase reactants, biomarkers such as CRP, are increased during these acute exacerbations. suPAR levels were measured in 43 patients with an acute exacerbation of COPD and compared to 30 healthy controls on day one of admission and seven days after treatment. All acute-phase reactants studied, including suPAR, and the clinically used CRP and fibrinogen, were markedly elevated. The AUC was superior for suPAR when compared to the other acute-phase reactants. The researchers concluded that suPAR can be used as a predictor for acute COPD exacerbations and in monitoring response to treatments [108]. A later meta-analysis of 11 studies involving 4520 patients confirmed these findings, with the additional finding that suPAR serves as a clinically useful biomarker for early diagnosis, with a sensitivity and specificity of 87% and 79%, respectively, and an AUC of 84%. These researchers also concluded that suPAR can be used to distinguish acute exacerbations from stable COPD, and also can be used to guide clinical response to treatment [109].

#### 2.11.5. Pleural and Parenchymal Acute Lung Injury and Repair

In the context of lung injury, uPAR has been associated with fibrotic processes and tissue remodeling. The uPA/uPAR complex activates plasmin and subsequently MMPs and other proteolytic enzymes, influencing the balance between tissue repair and fibrosis [175]. A correlation between the pleural fluid level of suPAR and the predictive requirement for aggressive management has been reported in patients with parapneumonic pleural effusions [120]. Clinical studies of 93 patients with pleural effusions found that levels of suPAR in the pleural fluid were significantly elevated, by even more than pH, glucose, and lactate dehydrogenase (LDH) [119]. In the same study, monitoring suPAR levels was the most accurate indicator for chest tube insertion. Additional clinical trials, involving more patients at multiple centers, are required to address this need for accurate biomarkers in clinical settings.

In mouse studies, diminished uPAR expression attenuated lung injury associated with hypoxia, and additionally, decreased lung parenchymal destruction [176]. Similarly, such anti-inflammatory effects limited the containment of pneumonia, which led to poorer outcomes.

The uPA/UPAR system has also been implicated in pulmonary fibrosis [177]. Shetty and colleagues reported that lung samples of patients with idiopathic pulmonary fibrosis (IPF) have increased uPAR expression compared to healthy patients. The same authors hypothesized that post-transcriptional regulation of uPAR plays a partial role in the development of lung fibrosis [178,179].

#### 2.11.6. Smoking Exposure

In a randomized controlled cohort study of 48 smokers vs. 46 never smokers, suPAR levels were found to be significantly elevated at 3.2 ng/mL vs. 1.9 ng/dL, respectively. Smokers were randomized to smoking cessation with nicotine patches alone. Four weeks following cessation, suPAR levels were comparable to never smokers. Individuals with the highest levels of smoking at the time of cessation maintained the highest levels at four weeks following cessation, pointing towards the utility of suPAR levels in CI. The researchers concluded that suPAR may be useful as a biomarker for personalization of smoking cessation by identifying those at risk and those who may benefit the most from cessation. However, future studies investigating the longitudinal effects are still needed to truly assess clinical utility [110].

### 2.12. Rheumatology

Rheumatoid arthritis and lupus are associated with altered uPAR activity (Table 1).

#### 2.12.1. Rheumatoid Arthritis

As mentioned in the inflammation subsection, suPAR levels in patients with rheumatoid arthritis are elevated when compared to healthy cohorts. suPAR levels correlated with joint involvement. These findings indicate that suPAR provides a useful biomarker for the diagnosis and management of progression of RA [85,86,160,161,162]. Potential treatments utilizing this biomarker are described in Section 3.

#### 2.12.2. Systemic Lupus Erythematosus (SLE)

SuPAR was proposed by rheumatology experts as a clinically useful biomarker for systemic lupus erythematous (SLE) as early as 2015 [180]. suPAR levels have been demonstrated to reflect accrued damage in patients with SLE. In a later study, researchers investigated suPAR levels in 344 patients with SLE who met the 1997 American College of Rheumatology classification criteria and compared these levels with organ damage as assessed by the SLICC/ACR Damage Index (SDI), a clinically used index of overall organ damage attributable to SLE. suPAR levels were elevated in patients with SLE with progressive damage when compared to those with no detected damage, in particular with those carrying an SDI of two or greater. Furthermore, in an optimized logistic regression analysis predicting damage attributable to SLE suPAR was identified as a useful biomarker for predicting disease progression, together with baseline disease activity (as identified by used SLEDAI-2K), age, and non-Caucasian ethnicity, with an AUC of 0.77, on the high side of a fair prognostication for organ damage accrual during the first five years of SLE disease [87].

#### 2.12.3. Systemic Sclerosis

Systemic sclerosis (SSc) is a multisystem autoimmune disease characterized by internal organ and dermal fibrosis, vasculopathy, and widespread immune system dysregulation. RA is likened to SSc in that they are both autoimmune diseases involving both the fibrinolytic and uPAR systems. SSc differs in that SSc is pathologically anti-angiogenic, whereas RA is pathologically pro-angiogenic [181,182]. suPAR levels were found to be elevated in patients with SSc versus controls, and those with pulmonary fibrosis had higher levels [183,184]. Mechanistically, the anti-angiogenetic properties of SSc in these endothelial cell types is chiefly due to uPAR diminishment and the resultant loss of the beta2 integrin-facilitated connection of uPAR with the actin cytoskeleton [185]. This cleavage is due to overproduction of matrix metalloproteinases in SSc cleaving and activating the uPAR, reducing angiogenesis in the process [186].

## 3. Development of New Therapeutics—uPAR as a Therapeutic Target

In Section 3, we will discuss uPAR as a potential therapeutic target for treating severe inflammatory diseases, with a focus on the virus-derived immune-modulating proteins.

Small-molecule-, antibody-, and protein-based uPA and uPAR inhibitors have been investigated and several have been briefly discussed above in the individual sections (Table 1), illustrating the efficacy of drugs that modify uPAR expression in disease. uPAR has been used for targeted chemotherapy through a variety of mediums including monoclonal antibodies and nanogels for treating cancers. A virus-derived immune-modulating protein, Serp-1 is a protein that binds uPA and uPAR and has been tested in a wide array of animal models with demonstrated efficacy in reducing inflammation and disease progression (Table 2) [187,188]. Serp-1 activity is lost in uPAR-knockout mouse models of aortic transplantation [146,147] and with anti-uPAR antibody treatments in mouse wound healing studies [189]. In-depth reviews have been compiled for many of these new approaches, and thus, we have provided a broader overview. Given the recent prior reviews on uPAR-modifying treatments [189,190,191,192,193,194], we have provided an overview of these approaches together with a more in-depth review of the virus-derived immune-modulating serpin, Serp-1 (Table 2).

The data examining uPAR blockade as a treatment further supports a central role for uPA/uPAR in disease progression and provides newer therapeutic targets for treating disease.

### 3.1. Oncology

Given uPAR’s diminished expression in healthy homeostatic tissues when compared to cancers, the receptor has been targeted as a viable therapeutic option [26]. Due to its role in plasminogen activation and tissue remodeling, allowing cellular invasion and altering immune responses, uPAR provides an avenue for cancer invasion and metastasis. The uPA/uPAR complex increases MMP proteolytic activity and dissolution of ECM barriers, along with enhancing stromal angiogenesis in the tumor microenvironment (TME). uPA/uPAR at the leading edge of immune cells may also enhance tumor-associated stromal cells, further increasing cancer growth and/or invasion. Many in vitro and in vivo studies have proven impaired tumor progression, metastasis, and invasion when the proteolytic function of uPA/uPAR is either impaired or inhibited [16,24,195,196].

The negative prognostic value with uPAR stromal expression in multiple cancer types, including colon, breast, and pancreatic cancers, clearly highlights the therapeutic potential of aiming at stromal TME as an adjuvant anti-angiogenic, anticancer treatment [191,192,193,194]. uPAR’s role in the tumor stromal microenvironment, overexpression in non-homeostatic tissue, high expression in aggressive cancer subtypes of poor prognosis, along with the lack of obvious cancer phenotypes when uPAR is deficient, all suggest uPAR as a candidate in anti-tumor cytotoxic therapy. Targeting strategies using uPAR have been employed.

uPAR targeting is accomplished using uPA-derived peptides, monoclonal antibodies, and high-affinity receptor-binding fragments of uPA (containing the GFD). As mentioned previously, uPAR plays an important role in malignancy (tumor invasion) and metastasis of breast cancer [66,67,68,135,136]. Thus, the therapeutic potential of this system has been explored. A fully human antibody called 2G10 that effectively blocks uPA/uPAR interactions and has shown promise in treating aggressive triple-negative breast cancer (TNBC) in mouse models has recently been developed [71].

These uPAR-targeting ligands provide scaffolds for targeted delivery of cytotoxic drugs, including traditionally used anticancer agents, stromal-targeting oncolytic viruses, cytotoxic products, clinically used radioisotopes, photosensitizers, chimeric antigen receptor (CAR) T cells, and immunostimulators [16]. These approaches increase tumor specificity and improve intra-tumoral delivery to the tumor stromal microenvironment. Albeit it is in its early research stages, the utility of uPAR targeting may provide effective targeted drug invasion through the dense stroma, a chief mechanism behind anti-chemotherapeutic drug resistance. Few medications exist that can overcome drug resistance [197,198].

uPAR-targeting radionuclide therapy, PET-probes based on the high-affinity peptide AE105, have been synthesized and tested preclinically in human xenograft mouse studies and in two clinical studies evaluating uPAR PET studies in breast, bladder, and prostate cancer patients, with positive results. Although further preclinical validation and toxicity studies are required, the availability of human anti-uPAR-targeting constructs and their successful use as adjuvant imaging agents strongly support clinical translation to determine the therapeutic and prognostic utility of uPAR in the management of aggressive tumors, such as ovarian, prostate, and breast cancers. These studies have implications for future use as an imaging or treatment adjuvant. The full cytotoxic effect on human tumor lesions in xenograft mouse models is likely understated as it leaves the host stromal compartment essentially unharmed, where uPAR operates in the TME [19,195,199,200,201,202,203,204]. This is further corroborated by the fact that when the radioimmunotherapy is combined with a stromal-targeting recombinant anti-uPAR antibody, complete tumor regression occurs with triple-negative breast cancer (TNBC) human tumor xenografts in a metastatic mouse model [196,205].

Recombinant immuno- and ligand-targeted fusion toxins (IT and LT) is another approach to cytotoxic therapy designed to target uPAR. The cell-binding domain of the toxin/ligand is complexed with the tumor-targeting vehicles, providing a desired binding specificity [206]. First described utilizing diptheria cytotoxin A, the catalytic potency of the toxin moiety is enhanced, allowing a small number of molecules to be effectively delivered to the cytosol to kill cancer cells [207,208]. Internalization of the toxin is, however, required, unlike the radionuclide therapies. In preclinical studies, utilizing mono- and bi-specific activating transcription factor (ATF)-fusion recombinant diptheria toxins that target uPAR has been successful in treating human to mouse xenograft glioblastoma multiforme (GBM) and non-small-cell lung cancer models [208,209,210,211,212,213].

Utilizing the same LT therapy with Pseudomonas aeruginosa toxin in GBM subcutaneous and intracranial xenograft models has also proven beneficial [213]. Mouse models investigating the recombinant toxin’s use in head and neck squamous cell carcinoma have found similar success [213]. The therapy was deemed safe and effective in targeting sarcomas in preclinical studies, but future clinical studies will be required [214]. A toxin from the plant bacteria saporin has been studied in human bladder and triple-negative breast cancer cell lines and in bladder cancer SC xenograft models with success [215,216]. The repeated success amongst different tumor subtypes in a variety of different toxin studies has now validated the ability to target uPAR expression in the TME once stromal penetrance occurs, utilizing the toxin-aided vehicles as the means of internalization.

Antibodies targeting uPAR utilizing the 2G10 antibody have also been designed with preclinical success. Clinical studies are still needed, but these may have future therapeutic implications in aggressive triple-negative breast cancer, for which current medical therapies are limited with minimal efficacy due to the lack of molecular targets [196,205]. Following blockage of the uPAR with antisense oligonucleotides, a uPAR expression was strongly reduced against murine prostate cancer bony metastases [217]. This very therapy also worked successfully in preclinical studies, reducing the invasion of human cartilage in synovial fibroblasts in vitro [218].

uPAR-targeting drug-loaded iron oxide nanoparticles (NPs) also have the capacity to direct therapy to target sites when complexed with guiding moieties, synergistically amplifying anti-tumor chemotherapeutic response. Preclinical trials have shown promise in treating human breast, prostate, and lung cancer cell lines. The chemotherapeutic drugs involved include doxorubicin with curcumin, gemcitabine, cisplatin, noscapine, and paclitaxel [75,77,219,220,221,222,223,224,225].

While the direct therapeutic uPAR-targeting antibodies have had success, immune-checkpoint blockade and adoptive cell therapy via chimeric antigen receptor (CAR) T cells have recently emerged as a breakthrough in the treatment of malignant tumors including ovarian, glioblastoma, and melanoma in preclinical studies. uPAR targeting was reported to selectively induce immune-mediated clearance of non-homeostatic uPAR-positive cancer cells via antibody-recruiting small molecules (ARMs) in a metastatic glioblastoma mouse model [226,227].

Oncolytic virotherapy is a new and rapidly growing field in cancer therapeutics, with proven late-stage clinical studies. Presently, one FDA-approved oncolytic virus exists, a herpesvirus [228]. However, virotherapy has taken advantage of the oncolytic measles virus, where success was seen in MV targeting the stromal in breast and colon human-mouse xenograft preclinical studies [229,230,231,232,233].

The virus-derived serpin Serp-1 binds uPA and uPAR in human macrophage cell lines. One study in severe combined immunodeficient (SCID) mice demonstrated that Serp-1 produced from myxomyavirus reduced pancreatic xenograft growth with altered myeloid immune cell responses (Figure 3, Table 2) [81]. Myxoma virus is being studied as an oncolytic virus for treating cancers. Angiogenesis, the growth of new blood vessels, also increases tumor growth. In very early work, chorioallantoic membrane angiogenesis was reduced by Serp-1 in an angiogenic model [144].

Despite these many promising studies—targeting uPAR as a new approach to cancer treatment has demonstrated great potential—this area remains in development.

### 3.2. Inflammatory Bowel Disease

uPA inhibitors have been recently investigated as potential new therapeutics for IBD. One study detected a dramatic reduction in colitis in mice with experimental colitis when treated with uPAR inhibitors.

The authors in the study suggested uPA as a therapeutic target for patients with UC. A modified, PEGylated Serp-1 protein, PEGSerp-1, has also been examined recently in acute severe and chronic DSS-induced colitis models, demonstrating improved survival and reduced colon inflammation and damage with loss of crypt architecture (Table 2) [155].

### 3.3. Anti-Inflammatory Anti-Angiogenic Immunomodulation

While originally investigated for its anti-metastatic potential, UPARANT (UPAR antibody) has been researched as an anti-inflammatory drug to treat angiogenic inflammatory disease states in patients with ocular pathologies and complications of diabetes [234]. UPARANT was found to reduce VEGF-induced angiogenesis caused by oxygen-induced retinopathy in a dose-dependent fashion by diminishing VEGF, VEGF-receptor 2, and transcription factors regulating VEGF expression [235]. In a mouse study modeling choroidal neovascularization, intravitreal injection of UPARANT reduced leakage from the choroid and choroidal neovascularization area [235,236]. In mouse edema and peritonitis models, intraperitoneal injection of UPARANT reduced pro-inflammatory enzymes [121].

In a mouse diabetic nephropathy model, injection of UPARANT subcutaneously restored vascular membrane integrity, making the vascular membrane less permeable [237]. Subcutaneous injection of UPARANT reduced Mueller cell gliosis, upregulated inflammatory markers, and had anti-apoptotic effects, all of which improved retinal function, in a retinitis pigmentosa mouse model [238]. In a mouse model of rubeosis iridis with proliferative retinopathy, local intravitreal injection of UPARANT was found to be superior to anti-VEGF treatment. Systemic subcutaneous injection also reduced neovascularization in this study [239]. All these findings are highly suggestive for UPARANT being a co-occurring anti-inflammatory and anti-angiogenic immune-modulating therapeutic.

Purified Serp-1 protein is a highly effective anti-inflammatory and immunomodulator, as has been proven in a wide variety of animal models and a clinical trial [188,240]. Serp-1 carries out its potent effects through binding to uPA, tPA, plasmin, thrombin, factor Xa and complement; Serp-1 requires uPAR for this potent immunomodulatory and anti-inflammatory activity. Upon binding, Serp-1 modulates monocyte and T cell cellular migration through uPAR-linked integrins and actin-binding proteins in Serp-1 cellular responses. Serp-1 interacts with uPAR to reduce macrophage activation, adhesion, and invasion. Serp-1 similarly diminished monocyte and T cell migration across endothelial layers and ascitic fluid both in vitro and in vivo [188].

Serp-1 increased expression of filamin B, an actin-binding protein and decreased beta-integrin (CD18) expression. These molecular functions were dependent upon uPAR, as demonstrated by immunoprecipitation analysis. Blocking Serp-1 induced changes in filamin B expression through small inhibitory RNA (siRNA)-reduced Serp-1-mediated inhibition of monocyte adhesion and diapedesis. In mouse models of aortic allograft transplantation, uPAR-deficient (uPAR-KO) donor allografts inhibited Serp-1-mediated reductions of aortic transplant vasculitis (Table 2). Antibodies to uPAR reduced the efficacy of Serp-1-mediated wound healing. The capacity of Serp-1 to reduce inflammation and associated diseases is, thus, closely associated with interactions with the uPA/uPAR complex, with Serp-1 treatment downregulating beta-integrin and increasing filamin B expression in human macrophage cell cultures. These findings further support the uPA/uPAR complex as a new therapeutic target [188,240].

Visual impairment due to photoreceptor degeneration in inherited eye diseases is chiefly due to inflammation, without angiopathy. uPA/uPAR dysfunction has been reported as a key player in a variety of eye diseases including retinitis pigmentosa and retinopathy of prematurity; emerging as a new inflammatory pathway in a variety of ocular diseases, a pathway that differs from the classic pro-angiogenic pathway. Through the counteraction of neovessel formation and microvascular dysfunction, uPAR modulates the inflammatory response as a possible treatment for a variety of eye diseases of neovascularity, including diabetic retinopathy, wet macular degeneration, retinopathy of prematurity, and retinitis pigmentosa. Finally, uPAR modulates the inflammatory cascade during rod-cell degeneration in retinitis pigmentosa. Retinitis pigmentosa is an eye disease with associated low uPA/uPAR levels. The uPA/uPAR complex may thus provide a possible therapeutic target through anti-inflammatory activity and reduction of oxidative stress [241].

In prior work, Serp-1 reduced alkali-induced corneal damage in a mouse model [233]. In a separate study, AAV (adeno-associated virus) expression of Serp-1 also reduced uveitis in vitro, corroborated with mouse models [124,145].

### 3.4. Diffuse Alveolar Hemorrhage in SLE

Diffuse alveolar hemorrhage (DAH) is a fatal complication of SLE with mortality of up to 50–80%. No proven effective treatments presently exist for DAH. In a mouse DAH model, treatment with PEGSerp-1 protein as well as the unmodified Serp-1 protein markedly reduced DAH, with associated reductions in uPAR distribution and reduced macrophage alveolar invasion (Table 2) [142,143]. In a collagen-induced arthritis model, early prophylactic treatment also reduced joint inflammation [154].

Future studies are needed to determine clinical significance; however, this finding again highlights the importance of uPAR in the processes of this potentially fatal autoimmune disease [142,143].

### 3.5. Acute Transplant Rejection

uPAR is linked to immune responses across the body including, but not limited to, the lungs and kidneys. Under conditions of oxidative stress and hypoxia the uPA/uPAR pathway is upregulated. Growing evidence has revealed that ischemia-reperfusion injury induces immune cell activation in both acute and chronic allograft rejection. Human biopsy studies have detected uPA/uPAR activation correlating with allograft rejection. The pathology contributing to acute allograft rejection is in part due to transplant vascular disease and occlusion and subsequent ischemia-induced cell damage and apoptosis. Recipient and also resident renal allograft leukocytes are reported to infiltrate the transplant organ. This is believed to be mediated in part through increased uPA/uPAR expression and dysregulation. uPAR has been found to be necessary for TNF-alpha and C5a signaling, inducing integrin ICAM-1 signaling on allograft endothelial cells, a crucial step in leukocyte diapedesis [242].

In prior work Serp-1 treatment improved outcomes in aortic allografts, renal allografts, heterotopic cardiac allografts, and rat-mouse cardiac xenografts (Table 2) [148,149]. As noted above, Serp-1 efficacy in aortic allograft models was blocked in uPAR-deficient allograft implants, demonstrating that Serp-1 efficacy is in part dependent upon uPAR.

### 3.6. Atherosclerotic Plaque Stabilization

Serp-1 has demonstrated preclinical efficacy in treating inflammatory atherosclerotic diseases in both preclinical and clinical studies (Table 2). In studies on hyperlipidemic rats and rabbits, continuous infusions of Serp-1, and later Serp-2, were found to reduce carotid cuff compression in ApoE null hyperlipidemic mouse models, stabilizing the carotid plaque at the affected sites [152].

In a randomized, blinded, dose-escalating phase 2A clinical trial at seven sites in the US and Canada, Tardif et al. demonstrated that Serp-1 given at the time of coronary stent implant for unstable coronary syndromes significantly reduced troponin and creatinine kinase MB (CK-MB) levels at the higher dose of 15 µg/kg. Serp-1 treatment reduced clinical markers of myocardial damage with no significant increases in adverse effects. Of interest, there were minimal if any detectable neutralizing antibodies to treatment with this virus-derived serpin protein [243]. The doses used were microgram/kilogram doses so the protein is highly active. All treatments with proteins and even antibodies can lead to antibody development, but the lack of neutralizing antibodies in response to this foreign protein is certainly promising for potential repeat treatments.

### 3.7. Inflammatory Vascular Disease

Giant cell arteritis is an inflammatory vascular disease with significant risks for sudden loss of vision (blindness) if left untreated, as well as other vascular complications including strokes, cardiomyopathy, and aneurysms. Serp-1 treatment successfully reduced inflammation in human giant cell arteritis xenograft implants in severe combined immunodeficient (SCID) mice [153] in the presence of peripheral human blood mononuclear cell infusions. The study of these temporal artery biopsy implants was blinded to GCA diagnosis. Gammaherpesvirus infection (MHV68) is also a model for inflammatory vasculitis [150]. Serp-1 treatment significantly improved survival and reduced vascular and lung inflammation in mouse models of MHV68 infection.

### 3.8. Inflammatory Arthritis

RA is the most common inflammatory rheumatologic disease in the world. In a collagen-induced arthritis model, rats administered with Serp-1 at 50 µg/kg via intravenous (IV) injections had reduced clinical arthritis, with reduced joint swelling when it was given at the time of inducing the disease and reduced bony erosions on radiographs, with overall improved clinical status [154].

### 3.9. Severe Acute Respiratory Distress Syndrome Due to SARS-CoV-2

Severe acute respiratory distress syndrome is a lethal sequalae of increased lung permeability, usually due to an infectious insult. In a mouse-adapted MA-30 SARS-CoV-2 model, SARS-CoV-2 treated with PEGSerp-1 altered uPAR gene expression and detectable protein on IHC analysis. Further treatment with PEGSerp-1 reduced lung and arterial inflammation and associated damage in the MA-30 SARS-CoV-2-infected C57Bl/6 mice [151]. As noted above, in prior work using an MHV68 mouse herpes model, causing an otherwise lethal infection, Serp-1 improved survival and reduced lung and vessel inflammation [149].

### 3.10. Inflammation in Wound Healing

Factors involving clotting, hemorrhaging, and inflammation help dictate early wound healing. In a mouse study, Serp-1 treatment in a chitosan–collagen hydrogel accelerated wound healing. This wound healing acceleration was blocked by the urokinase-type plasminogen activator (uPAR) antibody [156].

### 3.11. Neuromuscular Disorders, Duchene Muscular Dystrophy, and Spinal Cord Injury

Similar studies examined local infusions of Serp-1 as well as an implant of a chitosan–collagen hydrogel in a rat model of spinal cord injury. This treatment was compared to local dexamethasone treatments. Serp-1 improved motor function and reduced inflammation in this model without associated toxicity [157]. Mice deficient in dystrophin (*Dmd*) were administered PEGSerp-1 intraperitoneally for four weeks beginning at four weeks of age. These treatments reduced muscular inflammation and diaphragm fibrosis [158].

## 4. Discussion/Conclusions

In conclusion, within the realm of clinical applications, the uPA/uPAR complex is now proven to be a very promising biomarker for inflammatory disease and cancer. Noteworthy is the potential prognostic value in diverse clinical scenarios, specifically in cancer metastasis and severe inflammatory diseases such as viral infections and inflammatory bowel disease. A nuanced understanding of uPA/uPAR interactions, as presented in this review, is indispensable for the formulation of targeted therapies and interventions across various clinical domains.

The known benefits of using higher doses of uPA as a thrombolytic or clot-dissolving agent for acute thrombotic events, heart attacks, and strokes, where there is improved survival, appears to be at variance with the potential for damage from inflammation associated with the uPA/uPAR complex. However, the elevated uPA/uPAR complex response may have greater impact as part of an early inflammatory response. This early uPA/uPAR response may be initiated as an early healing response which, when excessive, then leads to more aggressive and damaging immune and inflammatory responses. The wide range of potential actions of uPA and suPAR may also provide differing responses to differing pathologies and/or, on a simpler level, differing cellular responses in differing organs. Further in-depth analyses of the local effects of the uPA/uPAR response in differing organs and cells may identify individual cell responses. An in-depth analysis will be of great interest and may provide new insights into these unique immune modifiers. The uPAR complex has a very wide range of functions in the inflammatory response, in addition to acting as a fibrinolytic. There are likely many new discoveries to be made in this field, both using uPAR as a marker for disease and as a new therapeutic target for treating diseases where treatments are limited.

Therapies targeting the uPA/uPAR complex from small molecules and antibodies, chemotherapy designed to target uPAR in cancer cells, and virus-derived immune-modulating protein such as the virus-derived serpin Serp-1 are under investigation and provide new approaches to treatment. The ongoing research and potential future directions, as discussed here, encompass the development of uPA/uPAR as diagnostic and prognostic tools and underscore a dynamic landscape of possibilities. In summation, this comprehensive review is intended to illuminate the diverse roles of uPA/uPAR and their clinical significance across multiple medical fields, setting the stage for novel insights and therapeutic avenues.

## Figures and Tables

**Figure 1 biomedicines-12-01167-f001:**
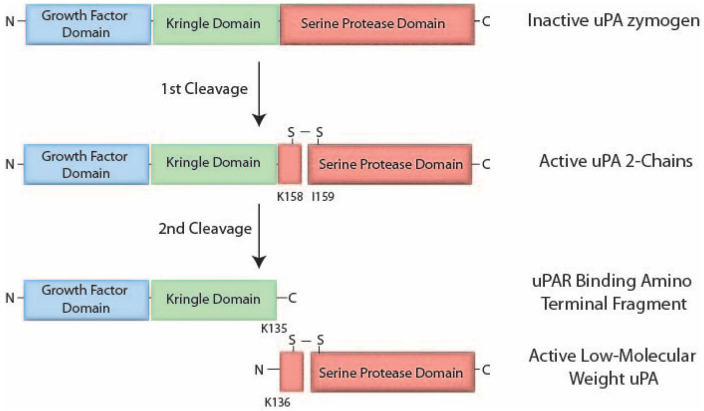
**Schematic of the enzymatic processing of uPA.** uPA is secreted as a 411-amino-acid-long inactive protein containing growth factor, kringle, and serine protease domains. Activation of uPA occurs through a proteolytic cleavage between K158 and I159 followed by linkage of the two peptides by a disulfide bond. A second round of proteolytic cleavage between K135 and K136 results in a catalytically inactive amino terminal fragment and a catalytically active soluble low-molecular-weight serine protease. uPA—urokinase-type plasminogen activator; uPAR—uPA receptor.

**Figure 2 biomedicines-12-01167-f002:**
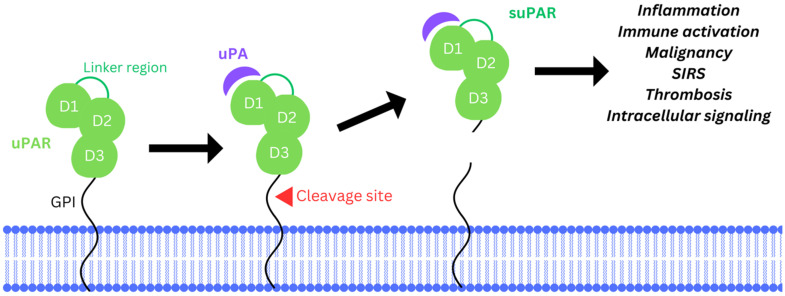
**Schematic of the uPA receptor (uPAR).** uPAR has three domains: D1, D2, and D3. D1 binds uPA and D3 is linked by GPI to the outer cell membrane. GPI—glycophosphatidyl inositol; uPA—urokinase-type plasminogen activator.

**Figure 3 biomedicines-12-01167-f003:**
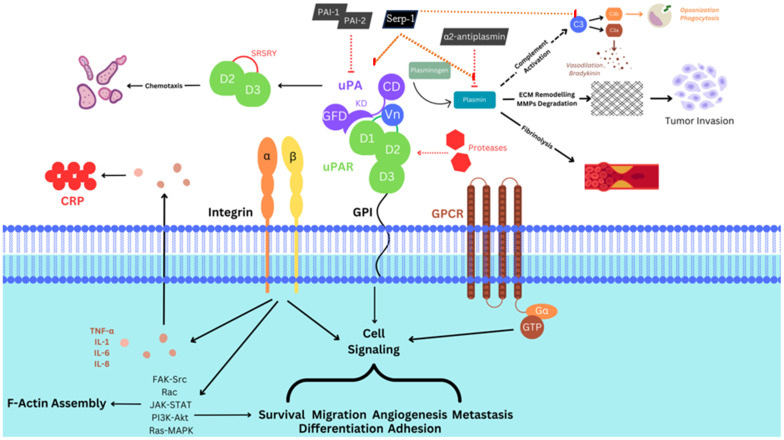
**The urokinase-type plasminogen activator receptor (uPAR) modifies intracellular and extracellular pathways.** uPAR is a GPI-linked membrane protein that modifies intracellular downstream pathway responses via protein-to-protein interactions in a large lipid raft of associated membrane proteins. uPAR modifies intracellular activation through interactions with integrins and GPCRs, as illustrated here. Mammalian serpin PAI-1 binding to the uPA/uPAR complex inhibits uPA/uPAR activity and can also be internalized, further modifying and altering intracellular activity pathways. uPA/uPAR also modifies extracellular activity via activation of plasminogen to form plasmin, with subsequent activation of MMPs and growth factors that alter cellular invasion into the extracellular matrix surrounding adjacent cells. The virus-derived serpin Serp-1 also binds and inhibits uPA and the uPAR as well as plasmin. MMP—matrix metalloproteinases; GPCR—G protein-coupled receptor; PAI—plasminogen activator inhibitor; C—complement; Vn—vitronectin; GPI—glycophosphatidyl inostitol; ECM—extracellular matrix; Serp-1—virus-derived serpin.

**Table 1 biomedicines-12-01167-t001:** uPAR/suPAR as diagnostic markers for clinical pathology.

Disease	Role of uPAR/suPAR	Target/Outcomes	Subjects	Citation(s)
Acute Coronary Syndrome and Percutaneous Coronary Intervention	(1) Increased uPAR on circulating monocytes in patients with ACS (2) Increased suPAR associated with increased MI and cardiac mortality (3) suPAR product of atherogenic cells	(1) uPAR levels are elevated in patients with ACS and myocardial necrosis. (2) uPAR predicts future MACE.	Human	[42,43,44,45]
Acute Myocardial Infarction (AMI)	(1) Deficient uPAR—protection against myocardial rupture and reduced myocardial infarct size (2) Reliable biomarker for mortality in adults with AMI and cardiac arrest	(1) uPAR-deficient mice protected against myocardial rupture with reduced infarct size. (2) suPAR levels predict 1 year mortality in AMI and correlate with neurological outcome and mortality after cardiac arrest.	Mice, Human	[46,47,48,49,50,51,52,53,54]
Coronary Artery Disease (CAD)	suPAR is a cleavage product of atherogenic immune cells	suPAR and uPAR levels correlate with extent of CAD.	Human	[47,55]
Cardiovascular Risk Stratification	suPAR is preditive of CV mortality	(1) suPAR, together with standard clinical markers, e.g., hs-CRP and NT-proBNP, is predictive of cardiovascular mortality. (2) suPAR outperforms hs-CRP for prognosis of inpatient mortality for coronary disease. (3) suPAR independent predictor for future adverse cardiac events.	Human	[56,57]
Viral Myocarditis	Remodeling of cardiac tissue with ventricular dysfunction	uPA-deficient mice with viral myocarditis protected against cardiac dilatation and failure.	Mice	[58]
Chronic Heart Failure	suPAR correlates with heart failure	suPAR concentration strongly correlates with mortality in patients with heart failure.	Human	[56]
Congenital Heart Block	Increased anti-Ro and uPAR associated with increased scar, collagen deposition, and heart block	uPAR colocalizes and interacts with Ro60 on apoptotic human fetal cardiomyocytes.	Human	[59,60]
Acute Ischemic Stroke	uPAR is linked to carotid vascular pathology	(1) Patients with higher uPAR have increased risk of ischemic stroke and carotid atherosclerosis. (2) uPAR levels in patients with carotid artery atherosclerosis are reduced with beta-blocker.	Human	[61,62,63]
Angiogenesis	The uPA/uPAR complex is directly involved in release of pro-angiogenic growth factors such as FGF-2 and VEGF	(1) Reduced uPAR in endothelial cells impairs VEGF signaling. (2) uPAR-deficent mice have incomplete angiogenesis.	Human Cells, Mice	[64,65,66]
Metastatic Cancer	uPAR is involved in extracellular matrix degradation, tumor angiogenesis, cell migration and proliferation, apoptosis, and contributes to multidrug resistance (MDR) in multiple cancers, specifically, breast, lung, prostate, head and neck, and ovarian cancers	Increased uPAR expression is consistently observed in various metastatic solid tumor tissues.	Humans	[67,68,69,70]
Breast Cancer		(1) Malignant breast tumors exhibit slightly higher uPA and PAI-1 mRNA expression but significantly increased uPAR mRNA expression compared to benign tissue. (2) HER-2 (+) breast cancer has higher uPAR expression.	Humans	[68,69,71]
Lung Cancer		(1) Monoclonal antibodies targeting uPAR decreased the invasive potential of lung cancer strain 95D cells in vitro. (2) uPAR-targeting U11 peptide conjugated with pH-sensitive doxorubicin and curcumin has synergistic anti-tumor effects on lung tumor cells.	In vitro, Mice	[72,73,74,75]
Prostate Cancer		(1) Inhibition of uPAR via MIR143 in nanoparticles inhibits tumor growth. (2) uPAR is highly expressed in non-homeostatic prostate tissue	Human Cell Lines	[70,76,77,78]
Head and Neck Squamous Cell Carcinoma (HNSCC)		uPAR is associated with poor prognosis and resistance to anticancer agents.	Human Cells, Clinical Samples	[79]
Ovarian Cancer		uPAR expression increases with lysophosphatidic acid (LPA) stimulation in ovarian cancer.	Human Cell Lines	[80,81]
Prostate Cancer		uPAR is highly expressed in prostate cancer.	Human Xenograft SCID mice	[82]
Leukemia	uPAR mRNA variants play a specific role in the progression of AML	Transfection of uPAR 3′UTR modulates pro-tumoral factors and cell adhesion.	Human Cell Lines	[83]
Chronic Inflammation	suPAR is a cleavage product that is elevated in acute and chronic inflammatory diseases and comorbidities, specifically, RA, SLE, SIRS, sepsis, HIV, TB, pancreatitis, hepatitis, liver failure, diabetes, kidney damage, asthma, pneumonia, COPD, and smoking	suPAR levels correlate with chronic inflammation.	Human Studies	[31,55,56,57,84]
Rheumatoid Arthritis (RA)		(1) suPAR levels exhibit a direct correlation with the number of inflamed joints. (2) Elevated suPAR levels are observed in RA patients.	Humans	[85,86]
Systemic Lupus Erythematous (SLE)		suPAR levels are elevated and predict disease progression.	Humans	[87]
Systemic Inflammatory Response Syndrome (SIRS)		(1) Initial suPAR concentrations significantly higher in patients who died within 28 days. (2) suPAR—good biomarker for differentiating SIRS from sepsis.	Humans	[88]
Bacterial and Viral Sepsis		(1) suPAR is a biomarker for identifying adult bacterial sepsis. (2) suPAR outperforms hs-CRP for hospital mortality prognosis for bacterial septic shock. (3) suPAR is superior to other biomarkers for differentiating septic vs. non-septic neonates.	Humans	[88,89,90,91]
HIV		(1) uPAR expression increased on lymphocytes and monocytes after HIV infection. (2) uPAR levels correlate with prognosis of HIV-1 similar to CD4+ count and viral load.	Human Studies	[92,93]
Tuberculosis (TB)		(1) suPAR levels are elevated in active tuberculosis. (2) suPAR is marker for treatment efficacy and elimination of TB.	Human Studies	[94]
Acute Pancreatitis		(1) suPAR differentiates severe acute pancreatitis (SAP) with high diagnostic accuracy. (2) suPAR differentiates SAP from mild pancreatitis. (3) uPAR predicts inpatient mortality from SAP. (4) suPAR correlates with clinical scores and lab values.	Humans	[95]
Chronic Hepatitis Fibrosis Progression		(1) suPAR is fair biomarker for liver fibrosis in chronic HCV. (2) suPAR distinguishes early and advanced fibrosis. (3) suPAR distinguishes cirrhotic and non-cirrhotic patients.	Humans	[96,97,98]
Acute Decompensated Liver Failure (ADLF)		(1) suPAR levels > 14.4 ng/mL predict 28 day mortality in patients with ADLF. (2) Ascitic suPAR levels increased in bacterial peritonitis.	Humans	[99]
Type 1 Diabetes Mellitus (T1DM)		(1) suPAR is elevated across all patients with T1DM. (2) Patients with CVD carried a 2.5 times higher suPAR level, 2.7 times for patients with autonomic dysfunction, 3.8 times for albuminuria, and 2.5 times higher for stiff arterial walls.	Human	[100]
Type 2 Diabetes Mellitus (T2DM)		(1) Higher baseline suPAR levels showed a higher risk of microalbuminuria in patients at risk for T2DM or patients with diagnosed T2DM.	Human	[101]
Acute Kidney Injury (AKI)		(1) Patients undergoing coronary angiography, cardiac surgery, or admitted to ICU within the upper quartile suPAR levels had increased risk for AKI and death at 90 days across all cohorts. (2) Mice given suPAR and contrast had greater pathologic evidence of AKI. (3) Monoclonal-uPAR-antibody-treatment mice had reduced AKI.	Humans, Mice Overexpressing uPAR	[101,102]
Focal Segmental Glomerosclerosis (FSGS)		Identified optimal suPAR value for diagnosis of FSGS was 4.644 ng/mL; sensitivity and specificity of 0.91 and 0.91; AUC of 0.946 and may be used to differentiate FSGS from other glomerular diseases.	Humans	[103]
Asthma		(1) Patients who were readmitted to hospital due to an acute asthma exacerbation or died had higher suPAR and decreased eosinophil on admission. (2) Patients in the 4th quartile for suPAR levels or eosinophil counts < 150 cells/uL had an increase in readmission or mortality.	Human	[104]
Community-Acquired Pneumonia		(1) suPAR levels were significantly elevated in patients with CAP and correlate with Pneumonia Severity Index. (2) LPS expression increases suPAR levels in macrophages.	Mice, Human	[105]
Ventilator-Associated Pneumonia		(1) Plasma suPAR levels were increased on day of diagnosis (AUC 0.77, *p* = 0.01) and in deceased patients (AUC 0.79, *p* < 0.001) in patients diagnosed with VAP. (2) suPAR significantly increased in patients with VAP 3 days before definitive diagnosis.	Human	[106,107]
Chronic Obstructive Pulmonary Disease		(1) Correlation superior for suPAR when compared to other acute-phase reactants, including CRP and fibrinogen. (2) suPAR is a clinically useful biomarker for early COPD diagnosis: sensitivity and specificity of 87% and 79%. (3) suPAR predictor for acute COPD exacerbation and monitoring treatment response.	Human	[108,109]
Smoking Exposure		(1) suPAR levels significantly elevated in smokers at 3.2 ng/mL vs. 1.9 ng/dL, respectively. (2) Four weeks following cessation, suPAR levels are comparable to never smokers when compared to those who did not stop smoking.	Humans	[110]
Neurological Disorders	(1) suPAR levels are elevated in the CSF in distinct CNS pathologies (2) uPAR is upregulated in microglial cells during acute intracerebral LPS exposure	(1) suPAR identified in CSF of patients with HIV dementia. (2) suPAR superior specificity when compared to CRP for discriminating osteomyelitis and other neurodegenerative spinal diseases.	Mouse Model, Humans	[63,111,112,113,114,115]
COVID-19 (SARS-CoV-2) Pneumonia	Altered uPA/uPAR expression associated with severe COVID-19	(1) uPAR associated with hypoxia and pneumonia. (2) COVID-19 patients with low suPAR have lower mortality. (3) suPAR guided anakinra treatment provided survival benefit.	Human Studies, Mouse	[116]
Inflammatory Bowel Disease (IBD)	uPAR maintains the integrity of the intestinal epithelial barrier	(1) uPAR expression increases during active-gut-damage IBD. (2) uPAR suppresses EGFR-modulated repair and signaling.	Cell Lines, Mouse	[117,118]
Parenchymal Lung Injury and Repair	uPAR associated with lung fibrotic processes and tissue remodeling	suPAR correlates with aggressive management in pleural effusions.	Human, Mouse	[119,120]
Ocular Diseases	uPAR linked to inflammatory neovascular formation in ocular diseases, specifically, premature retinopathy, retinitis pigmentosa, and wet macular degeneration	uPA/uPAR is reported in disease progression.	Mouse Model	[121]
Allograft Transplant Rejection	(1) uPA/uPAR activation correlates with allograft rejection (2) uPAR is necessary for TNF-alpha and C5a signaling, inducing integrin ICAM-1 signaling on allograft endothelial cells for leukocyte diapedesis	Serp-1 efficacy in aortic allograft models was blocked in uPAR-deficient allograft implants.	Mice	[122,123,124]

**Table 2 biomedicines-12-01167-t002:** Analysis of Serp-1 and PEGSerp-1 treatments targeting uPAR.

Therapeutic	Inflammatory Disorder	Treatment Effect/Outcome	Subjects Studied	Reference
Serp-1	Atherosclerotic Plaque Acute Coronary Syndromes with Stent Implant	(1) Multicenter phase II Clinical Trial in USA and Canada. Mechanism extensively studied. Reduced markers of cardiac damage; MACE = 0. No neutralizing antibodies. (2) Preclinical—reduced intimal hyperplasia.	(1) Human clinical trial, randomized dose-escalating trial at 7 sites in Canada and US (2) Preclinical—rabbits	[142,143]
Serp-1	Angiogenesis	Serp-1 reduced angiogenesis in chorioallantoic membrane.	Chicken	[144]
Serp-1	Pancreatic Cancer	Treated pancreatic xenografts had decreased growth with altered myeloid cell responses.	SCID mice (severe combined immunodeficiency model)	[82]
Serp-1	Corneal Abrasion	Corneal wound healing was enhanced by reducing immune cell infiltration, fibrosis, and neovascularization.	Mice	[123]
Serp-1	Uveitis	Inflammation was decreased with AAV expression Serp-1. Given intraocularly.	Mice	[124,145]
PEGSerp-1	Diffuse Alveolar Hemorrhage	PEGSerp-1 reduced lung hemorrhage and inflammation—effective when given as a delayed treatment.	Mice	[142,143]
Serp-1	Aortic, Renal, and Cardiac Allograft Acute and Chronic Transplant Rejection and Vasculitis	Serp-1 reduces chronic renal and aortic allograft vascular inflammation. Serp-1 lost activity in uPAR-KO mouse aortic allografts.	Mice, rats	[146,147,148,149]
Serp-1	Lethal Viral Infection	Serp-1 reduces vascular inflammation in a lethal gammaherpes MHV68 infection.	Mice	[150]
PEGSerp-1	SARS-CoV-2 Infection	PEGSerp-1 reduced immune coagulopathic lung damage after SARS-CoV-2 infection and reduced mortality and vascular inflammation in gammaherpes-infected mice.	Mice	[151]
Serp-1	Hyperlipidemic Atherosclerotic Plaque	Serp-1 infusion by osmotic pump significantly reduced carotid plaque after carotid cuff injury.	Mice—ApoE^−/−^ mice with carotid cuff injury	[152]
Serp-1	Giant Cell Arteritis	Reduced inflammation in human xenograft giant cell arteritis biopsy implants in SCID mice with PBMC infusions.	Mice	[153]
Serp-1	Inflammatory Arthritis (Rheumatoid Arthritis)	Serp-1 reduced joint inflammation in collagen-induced arthritis.	Rabbits	[154]
PEGSerp-1	Inflammatory Colitis	Colon damage was reduced in dextran sulfate sodium-induced colitis	Mice	[155]
Serp-1	Inflammation in Wound Healing	Serp-1 given at sites of skin wounds improved healing rates in mice and reduced inflammation Activity blocked by anti-uPAR antibody.	Mice	[156]
Serp-1	Spinal Cord Injury	Local infusion of Serp-1 after spinal cord injury in rats reduced inflammation and neuronal damage.	Rats	[157]
PEGSerp-1	Duchene Muscular Dystrophy	PEGSerp-1 reduced inflammation in mouse diaphragm and skeletal musculature.	Mice	[158]

## Data Availability

No new data were created or analyzed in this study. Data sharing is not applicable to this article.

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
