# Peer review of "Urokinase-Type Plasminogen Activator Receptor (uPAR) in Inflammation and Disease: A Unique Inflammatory Pathway Activator"

_biomedicines, 2024, doi:10.3390/biomedicines12061167_

Round 1
Reviewer 1 Report
Comments and Suggestions for Authors
This review has a very interesting topic: the roles of urokinase-type plasminogen activator and its potential use as part of diagnostic and therapeutic tools. Although initially considered only a part of thrombolytic cascade, new roles emerge as an inflammatory mediator or marker for metastatic cancer.
I am very pleased with the logic, didactic and comprehensive way the authors construct this review. For that I congratulate them.
There are only minor issues to be addressed.
I know that tables should be inserted near the place where they are first cited in text, however I don’t think that the place for Table 2 (dedicated to therapeutic use) is in Section 2 (a section dedicated mostly to its role as marker in diagnosis).
Also, at the end of Discussion section, the Tables are inserted again (the legend of Table 1 and both legend and table for Table 2). However, the legends are slightly different than the previous versions inserted in text. Also the content of Table 2 is similar but different than the version inserted in section 2 (I presume that one is a working version). Please review and correct.
Reviewer 2 Report
Comments and Suggestions for Authors
UPAR review
In-depth and detailed review of the role of the uPA/uPAR complex, sUPAR and related molecules in numerous pathologies with research perspectives for each of them and opportunities for development of new treatments, with abundant bibliographic documentation.
Well written article, very (too?) long to be read piece by piece.
I have no specific comments, a few typing errors here and there:
Lane 431 “occlussive”
Lane 501 “matstatic”
Lane 631 “infeftions”
Lane 780 “amnd”
Lane 1188”iare”
Reviewer 3 Report
Comments and Suggestions for Authors
In the field of research on inflammatory processes and tumors, the review by Hamada et al. addresses a topic that is both highly interesting and constantly relevant. The writers have covered a broad spectrum of subjects while providing a general overview of each. However, I do need to make a few corrections and clarifications. First of all I highly recommend reading these two reviews:
The urokinase receptor system, a key regulator at the intersection between inflammation, immunity, and coagulation.
Del Rosso M, Margheri F, Serratì S, Chillà A, Laurenzana A, Fibbi G.Curr Pharm Des. 2011;17(19):1924-43. doi: 10.2174/138161211796718189.
The plasminogen activation system in inflammation.
Del Rosso M, Fibbi G, Pucci M, Margheri F, Serrati S.Front Biosci. 2008 May 1;13:4667-86. doi: 10.2741/3032.
Concerning the text:
Lane 133: The reference n. 23 is incorrect. The bibliography is not precise, so please review it in its entirety.
Furthermore, to the best of my knowledge, three researchers—Vassalli JD, Del Rosso M., and Blasi F.—discovered the uPAR receptor simultaneously in 1985:
A cellular binding site for the Mr 55,000 form of the human plasminogen activator, urokinase. Vassalli JD, Baccino D, Belin D.J Cell Biol. 1985 Jan;100(1):86-92. doi: 10.1083/jcb.100.1.86.
Receptors for plasminogen activator, urokinase, in normal and Rous sarcoma virus-transformed mouse fibroblasts Del Rosso M, Dini G, Fibbi G.. Cancer Res. 1985 Feb;45(2):630-6. PMID: 2981611.
Differentiation-enhanced binding of the amino-terminal fragment of human urokinase plasminogen activator to a specific receptor on U937 monocytes.
Stoppelli MP, Corti A, Soffientini A, Cassani G, Blasi F, Assoian RK.Proc Natl Acad Sci U S A. 1985 Aug;82(15):4939-43. doi: 10.1073/pnas.82.15.4939.
Lanes 135-144: bibliographical references are missing
Lanes 184-208: bibliographical references are missing
Lane 146: The D1 domain's ability to bind uPA is indicated, but the cleavage of the D1 domain—a crucial process for the receptor's proper operation—is not mentioned.
I suggest you also mention Systemic Sclerosis which represents the alter ego of rheumatoid arthritis in terms of antiangiogenesis, both pathologies in which the fibrinolytic system and uPAR in particular play a crucial role.
Regarding the paragraphs on angiogenesis and prostate cancer and rheumatoid arthritis, I suggest you read the following papers. I would like to underline that the addition of these works is not required, they only represent a starting point for improving your review, if you find better references I encourage you to add them.
Endothelial progenitor cell-dependent angiogenesis requires localization of the full-length form of uPAR in caveolae.
Margheri F, Chillà A, Laurenzana A, Serratì S, Mazzanti B, Saccardi R, Santosuosso M, Danza G, Sturli N, Rosati F, Magnelli L, Papucci L, Calorini L, Bianchini F, Del Rosso M, Fibbi G.Blood. 2011 Sep 29;118(13):3743-55. doi: 10.1182/blood-2011-02-338681. Epub 2011 Jul 29.
Domain 1 of the urokinase-type plasminogen activator receptor is required for its morphologic and functional, beta2 integrin-mediated connection with actin cytoskeleton in human microvascular endothelial cells: failure of association in systemic sclerosis endothelial cells.
Margheri F, Manetti M, Serratì S, Nosi D, Pucci M, Matucci-Cerinic M, Kahaleh B, Bazzichi L, Fibbi G, Ibba-Manneschi L, Del Rosso M.Arthritis Rheum. 2006 Dec;54(12):3926-38. doi: 10.1002/art.22263.
Matrix metalloproteinase 12-dependent cleavage of urokinase receptor in systemic sclerosis microvascular endothelial cells results in impaired angiogenesis.
D'Alessio S, Fibbi G, Cinelli M, Guiducci S, Del Rosso A, Margheri F, Serratì S, Pucci M, Kahaleh B, Fan P, Annunziato F, Cosmi L, Liotta F, Matucci-Cerinic M, Del Rosso M.Arthritis Rheum. 2004 Oct;50(10):3275-85. doi: 10.1002/art.20562.
Effects of blocking urokinase receptor signaling by antisense oligonucleotides in a mouse model of experimental prostate cancer bone metastases.
Margheri F, D'Alessio S, Serratí S, Pucci M, Annunziato F, Cosmi L, Liotta F, Angeli R, Angelucci A, Gravina GL, Rucci N, Bologna M, Teti A, Monia B, Fibbi G, Del Rosso M.Gene Ther. 2005 Apr;12(8):702-14. doi: 10.1038/sj.gt.3302456.
Reduction of in vitro invasion and in vivo cartilage degradation in a SCID mouse model by loss of function of the fibrinolytic system of rheumatoid arthritis synovial fibroblasts.
Serratì S, Margheri F, Chillà A, Neumann E, Müller-Ladner U, Benucci M, Fibbi G, Del Rosso M.Arthritis Rheum. 2011 Sep;63(9):2584-94. doi: 10.1002/art.30439.
